# Testing the Limits of Fine-Tuning for Improving Visual Cognition in Vision Language Models

Luca M. Schulze Buschoff [* 1]   Konstantinos Voudouris [* 1]   Elif Akata [1 2]
Matthias Bethge [2]   Joshua B. Tenenbaum [3]   Eric Schulz [1]

## Abstract

Pre-trained vision language models still fall short of human visual cognition. In an effort to improve visual cognition and align models with human behavior, we introduce visual stimuli and human judgments on visual cognition tasks, allowing us to systematically evaluate performance across cognitive domains under a consistent environment. We fine-tune models on ground truth data for intuitive physics and causal reasoning and find that this improves model performance in the respective fine-tuning domain. Furthermore, it can improve model alignment with human behavior. However, we find that task-specific fine-tuning does not contribute to robust human-like generalization to data with other visual characteristics or to tasks in other cognitive domains.

## 1. Introduction

One of the main goals of machine learning research is to build machines that think and behave like humans. To meet this goal, Lake et al. (2017) proposed that human-like machine learning models must be capable of reasoning about their physical and social environment and its causal structure. These capabilities are sometimes summarized as *intuitive theories*—the cognitive expectations humans and other animals have about their environment from early on in development that they use to behave adaptively.

In this paper, we focus on two classes of intuitive theory. *Intuitive physics* relates to the ability to predict and understand the physical properties and interactions of inanimate objects

(Battaglia et al., 2012; Piloto et al., 2022), an ability that is present very early in development and does not require extensive learning or experience (Baillargeon et al., 1995; Spelke, 1990; Spelke & Kinzler, 2007). *Causal reasoning* describes the ability to infer cause-effect relationships (Waldmann, 2017; Pearl, 2009). There is growing evidence that humans possess an intuitive capacity to infer and predict causal relationships (Griffiths & Tenenbaum, 2009), and that this ability emerges early in development (Kuhn, 2012; Sobel & Kirkham, 2006). In the psychology literature, intuitive physics and causal reasoning have been studied most prominently in their relation to visual cognition—investigating how humans and other animals reason about their physical environment and its causal structure through the visual inputs they receive.

Vision language models (VLMs), which receive visual and textual linguistic input and produce textual output, have received recent attention for their apparently sophisticated reasoning in visual and linguistic tasks (Liu et al., 2025). However, recent work has established that VLMs are still limited in their understanding of the physical world and its causal structure (Jin et al., 2023; Balazadeh et al., 2024), suggesting that they lack human-like intuitive physics and causal reasoning. While VLMs perform reasonably well on intuitive physics problems, such as predicting the stability of block towers, they do not show a good fit with human behavioral data. On tests of causal reasoning, such as predicting whether removing a block would cause a tower to fall, VLMs perform poorly and again do not fit well with human behavior (Schulze Buschoff et al., 2025). Beyond the domains of intuitive physics and causal reasoning, VLMs have also been shown to have a number of visual deficiencies, and they often struggle with simple visual tasks that would be trivial for a human observer (Rahmanzadehgervi et al., 2024; Schulze Buschoff et al., 2025; Balazadeh et al., 2024). VLMs are prone to hallucinations, where the corresponding output does not sensibly correspond to the input image (Li et al., 2023; Liu et al., 2024). Ullman (2024) shows that VLMs hallucinate visual illusions where there are none, if the visual stimuli resemble canonical illusions that were likely in their training data. Similarly, Zhang et al. (2023) show that while the general alignment to human per-

---

[*]Equal contribution   [1]Institute for Human-Centered AI, Helmholtz Munich, Oberschleißheim, Germany [2]University of Tübingen, Tübingen, Germany [3]Department of Brain and Cognitive Sciences, MIT, Massachusetts, USA. Correspondence to: Luca M. Schulze Buschoff <lucaschulzebuschoff@gmail.com>.

*Proceedings of the 42$^{nd}$ International Conference on Machine Learning*, Vancouver, Canada. PMLR 267, 2025. Copyright 2025 by the author(s).

ception is low, larger models are somewhat susceptible to the same visual illusions as humans. Additionally, VLMs are not adversarially robust and are therefore subject to manipulation of both textual and visual inputs (Zhao et al., 2024). Campbell et al. (2024) suggest that the failures of VLMs on tasks containing multiple objects can be explained by a *binding problem*, in which VLMs, like humans (Frankland et al., 2021), struggle to attend to, represent, and distinguish between multiple objects at the same time, because they share the same representational resources.

In pursuit of improving the performance of language models, *fine-tuning* is quickly distinguishing itself as the gold standard, enabling researchers to efficiently steer models towards better capabilities (Han et al., 2024) as well as towards more human-aligned outputs (Binz et al., 2024; Hussain et al., 2024). In this paper, we explore whether fine-tuning VLMs on single tasks can improve their performance on intuitive physics and causal reasoning tasks in the visual domain, as well as steer them towards more human-aligned outputs.

However, a hallmark of human cognition is not just the ability to reason about the physical environment and its causal structure, but also to robustly generalize from limited experience to solve new tasks (Collins et al., 2022; Geirhos et al., 2018; Griffiths & Tenenbaum, 2009). Therefore, we seek to evaluate whether task-specific fine-tuning not only improves performance on visual cognition tasks sampled from an identical distribution, but also whether it produces models that can generalize to new, but related, tasks in new domains. For example, we ask whether a model fine-tuned to accurately judge the stability of short tower blocks can generalize this knowledge to judge the stability of tall tower blocks, of tower blocks with different visual characteristics (from a different environment), or to causal reasoning problems about tower blocks. Our results allow us to appraise the limits of task-specific fine-tuning for building performant, human-like machine learning models that can generalize beyond the kinds of data on which they have been trained. Across a range of datasets and models, we do not find evidence that fine-tuning alone can achieve all these objectives.

### 1.1. Related Work

Closest to our work is Balazadeh et al. (2024), who fine-tune the VLM PaliGemma-3B (Beyer et al., 2024) on a series of intuitive physics and visual reasoning tasks, asking questions about the height, color, and shape of tower blocks in an image, as well as whether the towers are stable or certain blocks are likely to move. They find that smaller fine-tuned VLMs can outperform larger non fine-tuned models on the fine-tuning task. However, they do not investigate whether fine-tuned VLMs can generalize to new problems.

Ming & Li (2024) explore VLMs' ability to generalize to out-of-distribution labels in an image classification task, presenting evidence that fine-tuning noticeably improves performance. However, they do not investigate generalization in more complex, psychologically-inspired domains like intuitive physics or causal reasoning. Chen et al. (2021) find that a neurosymbolic (non-transformer-based) model can robustly reason causally about visual scenes in the CLEVRER dataset (Johnson et al., 2017; Yi et al., 2020), and generalize to new causal reasoning tasks. Generalization and causal reasoning has also been studied extensively outside of the visual cognition domain, such as mathematics (Zhou et al., 2023; 2024) and compositional reasoning (Dziri et al., 2023; Li et al., 2024). Binz et al. (2024) find that fine-tuning on diverse human behavioral data can confer an advantage on a wide range of tasks relevant to human psychology.

### 1.2. This Work

In this work, we fine-tune VLMs on single tasks from two cognitive domains inspired by research in cognitive science, intuitive physics and causal reasoning (Baillargeon & Hanko-Summers, 1990; Baillargeon et al., 1992; Battaglia et al., 2012; Lake et al., 2017; Lerer et al., 2016; Piloto et al., 2022; Spelke et al., 1992). In particular, we focus on model intuitions about the factual and counterfactual stability of stacks of coloured, uniformly dense blocks. We design these tasks in ThreeDWorld (Gan et al., 2020, TDW), a virtual environment with a realistic physics engine built in Unity (Unity Technologies, 2023). We refer to our dataset of block towers built in TDW as *Cubeworld*. We then evaluate the fine-tuned models' ability to generalize to four different conditions (see Figure 1):

1. A held-out test set randomly sampled from the same distribution as the fine-tuning data. *Example*: A model fine-tuned to judge the stability of towers consisting of 2–4 blocks is then tested on new unseen towers consisting of 2–4 blocks.
2. A test set of new block stacks, on the same task and domain as the fine-tuning data (e.g., tower stability). *Example*: A model fine-tuned on 2–4 block towers is tested on 5–7 block towers.
3. A test set of block stacks from the same task and domain but with different visual characteristics. *Example*: A model fine-tuned on 2–4 block towers from the *Cubeworld* environment is tested on real block towers with 2–4 blocks from Lerer et al. (2016).
4. A test set of block stacks from a new cognitive domain that shares the same visual characteristics. *Example*: A model fine-tuned to make stability judgments (intuitive physics) is tested on its ability to make counterfactual stability judgments (causal reasoning).

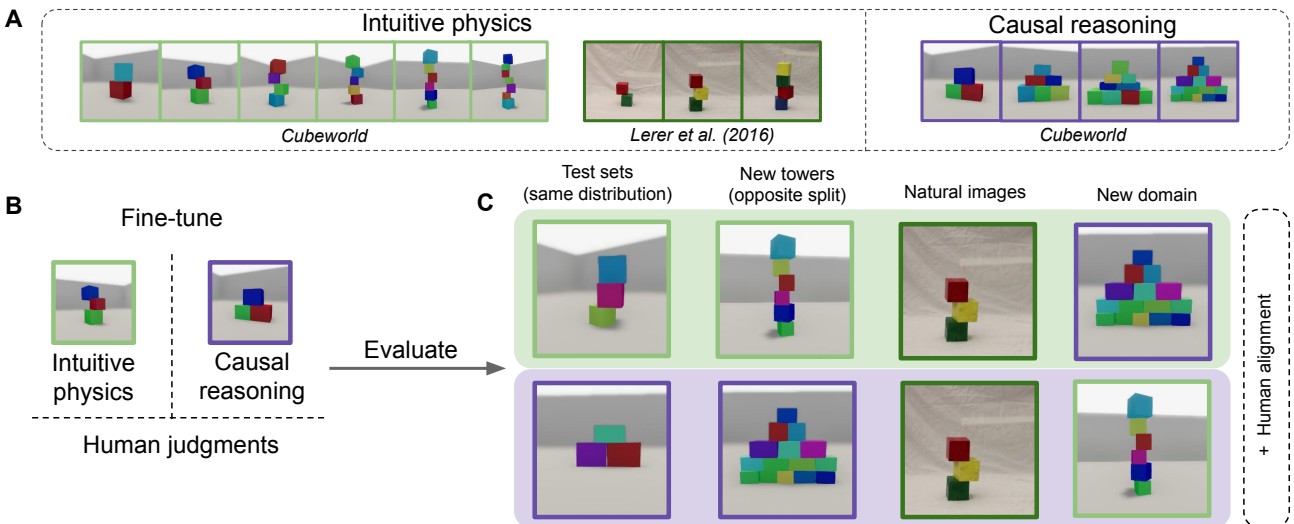

*Figure 1.* Methodology overview. **A:** We study causal reasoning and intuitive physics using our *Cubeworld* fine-tuning and evaluation datasets and the Lerer et al. (2016) block tower evaluation dataset. **B:** Models are fine-tuned on the ground truth or human judgments for a domain of the *Cubeworld* dataset. **C:** We test whether fine-tuning improves model performance in four scenarios: new towers of the same heights as in training; new towers of different heights compared to training; naturalistic images from Lerer et al. (2016); and block towers from the other domain. We also test the alignment of these models to human responses.

In each case, we not only measure how well the models perform in each context, but also how well their performance aligns with human data on identical tasks. We conduct counterbalanced evaluations, testing the interactions between tower sizes, visual characteristics, and cognitive domain (intuitive physics vs. causal reasoning). Finally, we fine-tune models on human judgments to test if this leads to better human alignment.

## 2. Methods

### 2.1. Fine-Tuning & Evaluation Data

We generated four new data sets, two data sets for intuitive physics, and two for causal reasoning. For each cognitive domain, we used one set for fine-tuning and one set for evaluation. All four data sets, which we collectively call *Cubeworld*, consist of different configurations of colored blocks.

We generated similar stimuli for both domains to ensure that models can, in theory, transfer knowledge between them. This allows us to test generalization within domains (such as fine-tuning models on physical stability judgments of small towers and testing on bigger towers) and between domains (such as fine-tuning models on physical stability judgments and testing them on counterfactual stability judgments).

**Intuitive physics** For intuitive physics, we generated block towers that consist of stacks of single colored blocks in a minimal gray room (see section A.1 in the Appendix). Block

towers such as these have been used extensively to investigate intuitive physics in humans and machines (Battaglia et al., 2013; Lerer et al., 2016).

The towers consist of 2 to 7 blocks and their rotation, size, color, and offset are sampled randomly. Offset distributions become more constrained as the number of blocks increases, so that randomly sampling offsets leads to a roughly 50/50 split between stable and unstable configurations for all tower sizes. This is to ensure that the distributions of stable and unstable towers have comparable difficulties: both contain easy *canonical* configurations as well as configurations that are harder to judge. The models are presented with an image of a block tower and they must judge if it is *stable* or *unstable*.

**Causal reasoning** For causal reasoning, we generated colored block pyramids in a minimal gray room, inspired by the stimuli in Zhou et al. (2022) (see section A.2 in the Appendix). The pyramids are made up of 2 to 5 rows with the bottom row consisting of as many blocks as there are rows in total, and each consecutive row featuring one less block than the row below (resulting in a range of 3 to 15 blocks in total). The color of each block is sampled randomly and the offset and sizes are sampled within ranges that still allow for a stable pyramid.

Each pyramid features a red block. The models are asked if any other blocks would fall if the red block were not there, similar to the the protocol of Zhou et al. (2022). This question requires the models perform the counterfactual

simulation of computing the stability of the tower without the red block. We randomly sample the position of the red block so that it is never on the top of the pyramid and so that it has an equal chance of being in every row of a pyramid.

**Naturalistic Data** To study whether models could generalize to data with other visual characteristics, we used a sample of 100 intuitive physics tower block images from Lerer et al. (2016) (see section A.3 in the Appendix). This dataset consists of pictures of real block towers with 2, 3, and 4 blocks that are either stable or unstable. The images look different to *Cubeworld*, but the underlying cognitive task is the same as in the intuitive physics data set. Human data for this task was taken from Schulze Buschoff et al. (2025), who collected 107 participants on 100 randomly selected images from the experiment by Lerer et al. (2016).

## 2.2. Models

We fine-tune the 7B parameter version of the Qwen2-VL model (Wang et al., 2024) and the 11B and 90B versions of Llama 3.2 (Grattafiori et al., 2024) using the *unsloth* library (Han et al., 2023). We used pre-trained models quantized to 4-bit precision.

We evaluate the models by sampling the log probabilities of the "Yes" and "No" tokens conditional on the input and normalizing them by exponentiating and then using the softmax function.[1] This gives a measure of the relative probability of the model answering "Yes" or "No" to each question. We then evaluate whether the model is correct by examining which token is assigned the higher probability and comparing this to the ground truth (see section C in the Appendix for information on the packages used for analysis). We also elicited free text responses from the model and found that these aligned with the normalized token probabilities anyway.

## 2.3. Prompts

For intuitive physics, we prompt the models with the following pre-prompt: "*You are now viewing a tower of blocks. Will the tower fall? Answer Yes if you think this tower is unstable and will fall. Answer No if you think this tower is stable and will not fall.*"

For the causal reasoning pyramids, we prompt the models with this pre-prompt: "*You are now viewing a pyramid of blocks. If the red block was not there, would any other blocks fall? Answer Yes if you think that other blocks would fall if the red block was not there. Answer No if you think that no other blocks would fall if the red block was not there.*"

---

[1]$\text{softmax}(\mathbf{p}) = \frac{e^{p_i}}{\sum_j^K e^{p_j}}$ where $\mathbf{p}$ is the vector of probabilities of length $K$ and $e$ is the exponential function.

## 2.4. Fine-Tuning Procedure

We used Parameter Efficient Fine-Tuning (PEFT; Han et al., 2024), focusing on training low-rank adapters for quantized models (QLoRA; Dettmers et al., 2024; Hu et al., 2021). PEFT is quickly becoming the dominant fine-tuning paradigm, blending high performance with computational and memory efficiency. PEFT selectively adjusts only a small number of model parameters during training, which not only reduces computational overhead but also minimizes overfitting and the prospect of existing knowledge being washed out by subsequent training (catastrophic forgetting; French, 1999). QLoRA is an approach to PEFT where the model is first quantized, reducing its memory footprint by reducing the precision of the models weights and activations, and then injecting small *adapter* layers into the transformer blocks of the VLM, both for the vision encoder and the autoregressive text decoder. For the weight matrix, $W$, of any layer, an accompanying adapter layer, $W_a$, is injected. $W_a$ is the product of two low-rank matrices $L_1$ and $L_2$ where $L_1 \in \mathbb{R}^{d \times r}$ and $L_2 \in \mathbb{R}^{r \times k}$ where $r$ is much smaller than $d$ and $k$, the dimensionality of the input and output and respectively. Given some input $x$, it is transformed by both $W$ and $W_a$ independently and then summed (subject to scaling $\alpha$), to produce the output. In QLoRA, only the values of $L_1$ and $L_2$ are altered; $W$ remains fixed. The weights of $L_1$ and $L_2$ are altered by backpropagation under the supervision of the next token in a document, using a cross-entropy loss. Given the relatively small $r$, models can be trained much more quickly than through training the full-rank $W$ matrix. We chose $r = 16$ for all experiments and a scaling of $\frac{r}{\alpha}$ where $\alpha = 16$, thus balancing the effect of $W$ and $W_a$ on the outputs. We fine-tuned layers in the ViT vision encoder, and attention and MLP layers in the language decoder, as this has been shown to be effective in prior work on fine-tuning for intuitive physics understanding (Balazadeh et al., 2024). We used the ADAM optimizer and an initial learning rate of 0.0002. We fine-tuned all models for 10 epochs on 10,000 text-image pairs on 80GB NVIDIA A100 GPUs. To ensure the robustness of our results, we repeated every experiment with three different seeds, leading to different samples of training data and different adapter-weight initializations, and report all results as averages across the three repeats.

## 2.5. Human experiments

We performed three separate human experiments to obtain fine-tuning and evaluation data. All participants agreed to take part in the study and were informed about the general purpose of the experiment. Experiments were conducted on Prolific in accordance with the relevant guidelines and regulations approved by the ethics committee of the University of Tübingen. For information on the samples, durations, and payout of the experiments, see Section B in the Appendix.

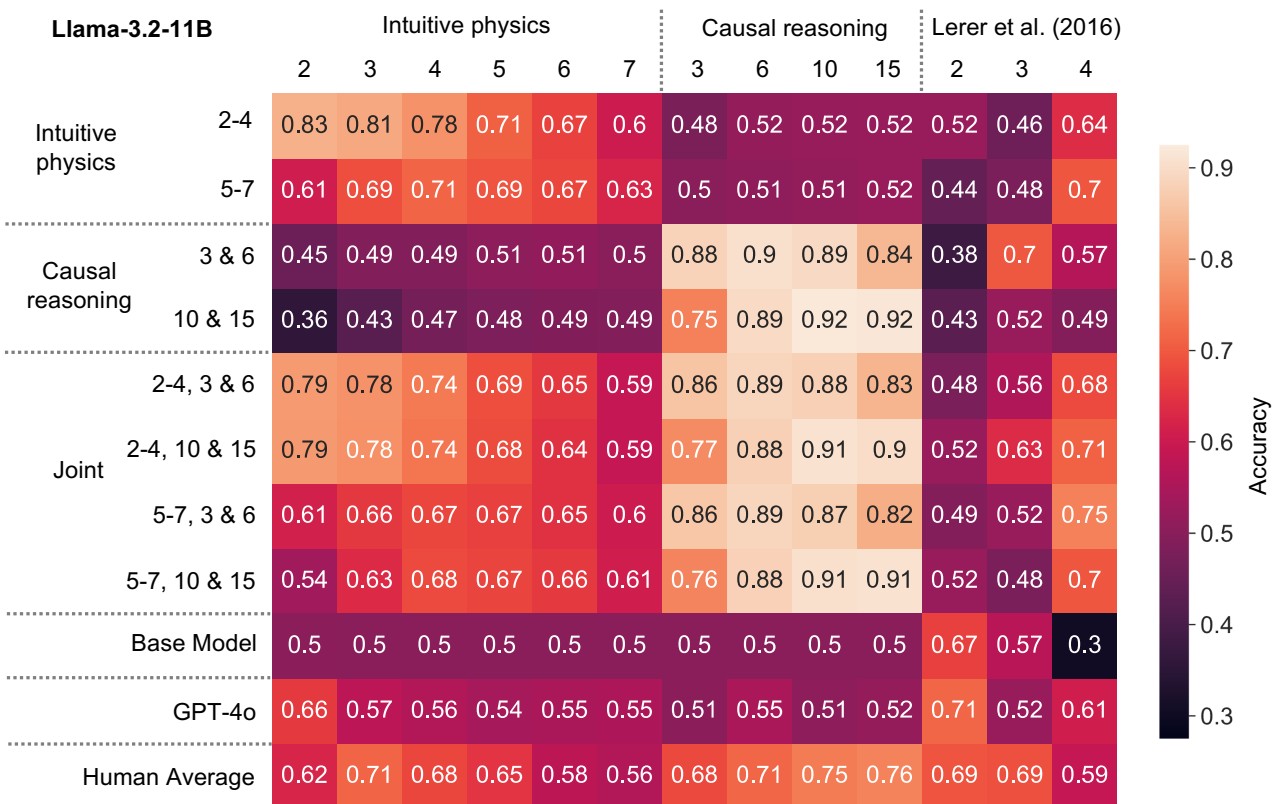

| Llama-3.2-11B | | Intuitive physics | | | | | | Causal reasoning | | | | Lerer et al. (2016) | | |
|---|---|---|---|---|---|---|---|---|---|---|---|---|---|---|
| | | 2 | 3 | 4 | 5 | 6 | 7 | 3 | 6 | 10 | 15 | 2 | 3 | 4 |
| Intuitive physics | 2-4 | 0.83 | 0.81 | 0.78 | 0.71 | 0.67 | 0.6 | 0.48 | 0.52 | 0.52 | 0.52 | 0.52 | 0.46 | 0.64 |
| | 5-7 | 0.61 | 0.69 | 0.71 | 0.69 | 0.67 | 0.63 | 0.5 | 0.51 | 0.51 | 0.52 | 0.44 | 0.48 | 0.7 |
| Causal reasoning | 3 & 6 | 0.45 | 0.49 | 0.49 | 0.51 | 0.51 | 0.5 | 0.88 | 0.9 | 0.89 | 0.84 | 0.38 | 0.7 | 0.57 |
| | 10 & 15 | 0.36 | 0.43 | 0.47 | 0.48 | 0.49 | 0.49 | 0.75 | 0.89 | 0.92 | 0.92 | 0.43 | 0.52 | 0.49 |
| Joint | 2-4, 3 & 6 | 0.79 | 0.78 | 0.74 | 0.69 | 0.65 | 0.59 | 0.86 | 0.89 | 0.88 | 0.83 | 0.48 | 0.56 | 0.68 |
| | 2-4, 10 & 15 | 0.79 | 0.78 | 0.74 | 0.68 | 0.64 | 0.59 | 0.77 | 0.88 | 0.91 | 0.9 | 0.52 | 0.63 | 0.71 |
| | 5-7, 3 & 6 | 0.61 | 0.66 | 0.67 | 0.67 | 0.65 | 0.6 | 0.86 | 0.89 | 0.87 | 0.82 | 0.49 | 0.52 | 0.75 |
| | 5-7, 10 & 15 | 0.54 | 0.63 | 0.68 | 0.67 | 0.66 | 0.61 | 0.76 | 0.88 | 0.91 | 0.91 | 0.52 | 0.48 | 0.7 |
| Base Model | | 0.5 | 0.5 | 0.5 | 0.5 | 0.5 | 0.5 | 0.5 | 0.5 | 0.5 | 0.5 | 0.67 | 0.57 | 0.3 |
| GPT-4o | | 0.66 | 0.57 | 0.56 | 0.54 | 0.55 | 0.55 | 0.51 | 0.55 | 0.51 | 0.52 | 0.71 | 0.52 | 0.61 |
| Human Average | | 0.62 | 0.71 | 0.68 | 0.65 | 0.58 | 0.56 | 0.68 | 0.71 | 0.75 | 0.76 | 0.69 | 0.69 | 0.59 |

*Figure 2.* Heat map showing accuracies for the 11B model on all combinations of ground truth fine-tuned models and evaluations. Each row contains a model fine-tuned on the ground truth for a specific data split. Each column contains the results for a specific block number in each evaluation data set. Models fine-tuned on a single cognitive domain do not generalize to the other cognitive domain. Models fine-tuned on both cognitive domains perform well on both domains as well as on the naturalistic Lerer et al. (2016) dataset.

**Intuitive physics** For the majority of results reported here, the model is fine-tuned on the ground-truth of the generated block configuration. However, we also fine-tune the models on human responses. For this purpose, we collected individual human responses for each image in the 2–4 block tower intuitive physics fine-tuning data set. We collected 100 responses on average from 100 human participants to cover the 10,000 images in the fine-tuning data set. In this experiment, all participants received different images and were given the same pre-prompt as the models in the intuitive physics experiment (see Section 2.3).

We also collected the responses of 100 separate participants on the same subset of 120 images from all conditions in the evaluation set for the intuitive physics tower task (6 tower sizes × stable / unstable × 10 images per condition). This allows us to compute similarities between human and model judgments. Participants received the images in a random order and were given the same prompt as in the experiment above.

**Causal reasoning** For the causal reasoning pyramids, we also collected a human evaluation data set of 100 separate participants on the same 80 images in the evaluation set (4

pyramid sizes × stable / unstable × 10 images per condition). Participants received the images in a random order and were given the same pre-prompt as the models in the causal reasoning experiment (see Section 2.3).

## 3. Results

First we fine-tune on the ground truth. We evaluate whether this leads to improved performance (3.1), different types of generalization (3.2–3.4) and alignment to human judgments (3.5). We then fine-tune a model on human responses on the same task (3.6), which leads to better human alignment. All results reported here are averaged over three seeds. The random seed changes the random initialisation of the fine-tuning weights and subset of the fine-tuning data.

### 3.1. Fine-tuning performance improvement

Fine-tuning substantially improves the performance of most models compared to the zero-shot case. Fully fine-tuned 11B models achieve accuracies between 0.6 and 0.92 on single block sizes from the split they were fine-tuned on (see Fig. 2), compared to the zero-shot base models, which

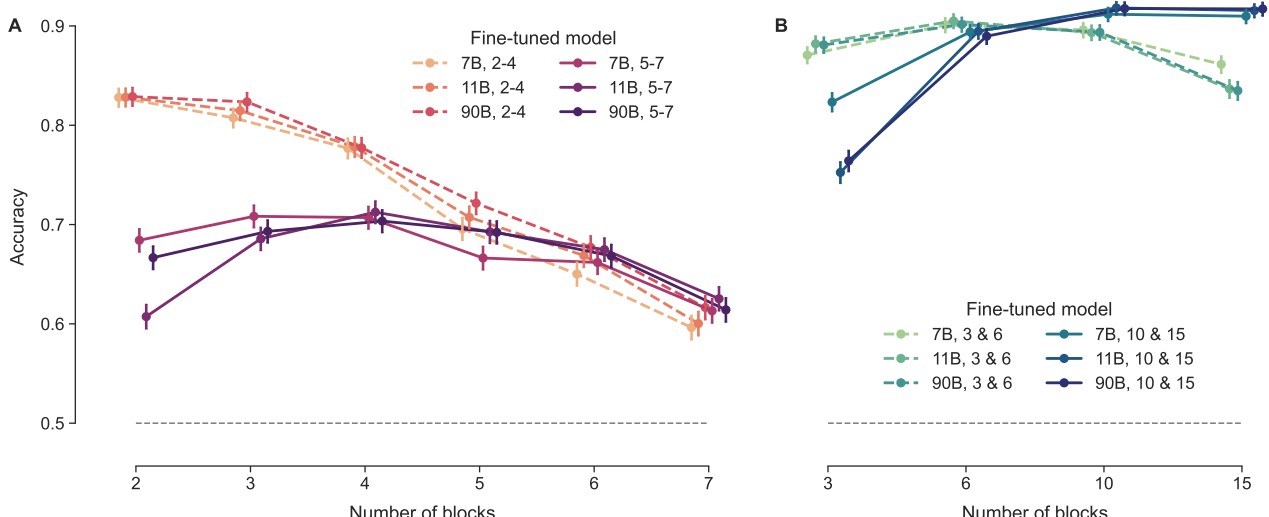

*Figure 3.* **A:** Models fine-tuned on two splits of intuitive physics towers (2–4 or 5–7 blocks) and evaluated on all tower sizes. Performance for models fine-tuned on 2–4 block towers decreases with tower size. Models fine-tuned on 5–7 block towers show similar performance over all tower sizes. **B:** Models fine-tuned on two splits of causal reasoning pyramids (3 & 6 or 10 & 15 blocks) and evaluated on all pyramids sizes. All models perform better on pyramid sizes they have been trained on.

perform at around chance for all tower and pyramid sizes (see Figs. 11B and 12B in Appendix E).

For the intuitive physics fine-tuned models, we find that the 7B, 11B and 90B models fine-tuned on 2–4 block towers achieve accuracies between 0.78 and 0.83 on towers from their fine-tuning distribution (see Fig. 3A). This picture is not as clear for the models fine-tuned on 5–7 block towers, with all models showing more or less similar performance improvements over all tower sizes.

The models might have difficulty learning from the 5–7 block towers because judging the stability of a tower becomes harder as it increases in size. This is mirrored in human performance on the evaluation data set, with human average accuracies of 0.62, 0.71, and 0.68 for towers of size 2, 3, and 4, and average accuracies of 0.65, 0.58, and 0.56 for towers of size 5, 6, and 7. The mean human accuracy over all towers was 0.63.

We find that the models fine-tuned on causal reasoning improve in performance on all pyramid sizes regardless of their fine-tuning split (see Fig. 3B). This is likely because the causal reasoning data set is easier to learn. Human participants had an average accuracy of 0.72, with accuracies of 0.68 and 72 for 3 and 6 block pyramids, and accuracies of 0.75 and 0.76 for 10 and 15 block pyramids.

### 3.2. Generalizing to taller and shorter towers

Models are able to generalize to taller and shorter towers to some degree. For models fine-tuned on 2–4 block intuitive physics towers, we see that they are able to somewhat generalize to bigger towers (see Fig. 3A). While their per-

formance decreases as the number of blocks increases, it is still above that of the base model even for bigger towers.

In contrast, the models fine-tuned on 5–7 towers do not show a strong difference in performance between towers that were in- and out-of their fine-tuning distribution. Crucially, they only performs as well on the 5–7 block towers as the 2–4 fine-tuned models, even though these latter models have to generalize from their fine-tuning distribution to bigger towers.

The performance of the causal reasoning fine-tuned models is more constant over different pyramid sizes (see Fig. 3B). Still, models fine-tuned on 10 & 15 block pyramids perform slightly worse on 3 block pyramids (see Fig. 11A).

### 3.3. Generalizing to a different visual quality

Models fine-tuned on artificial block towers do not generalize well to realistic block towers. To ascertain to what extent fine-tuned models can generalize to the same task with different visual characteristics, we tested them on real images depicting block towers from Lerer et al. (2016). We find that models fine-tuned on a single domain do not generalize well to all tower sizes in the Lerer et al. (2016) dataset (see Fig. 2). For example the 11B model fine-tuned on 2–4 block towers in *Cubeworld*, which is identical to the Lerer data in task and the number of blocks, only performs above chance on towers with 4 blocks from the Lerer dataset (see also Fig. 13 in Appendix F).

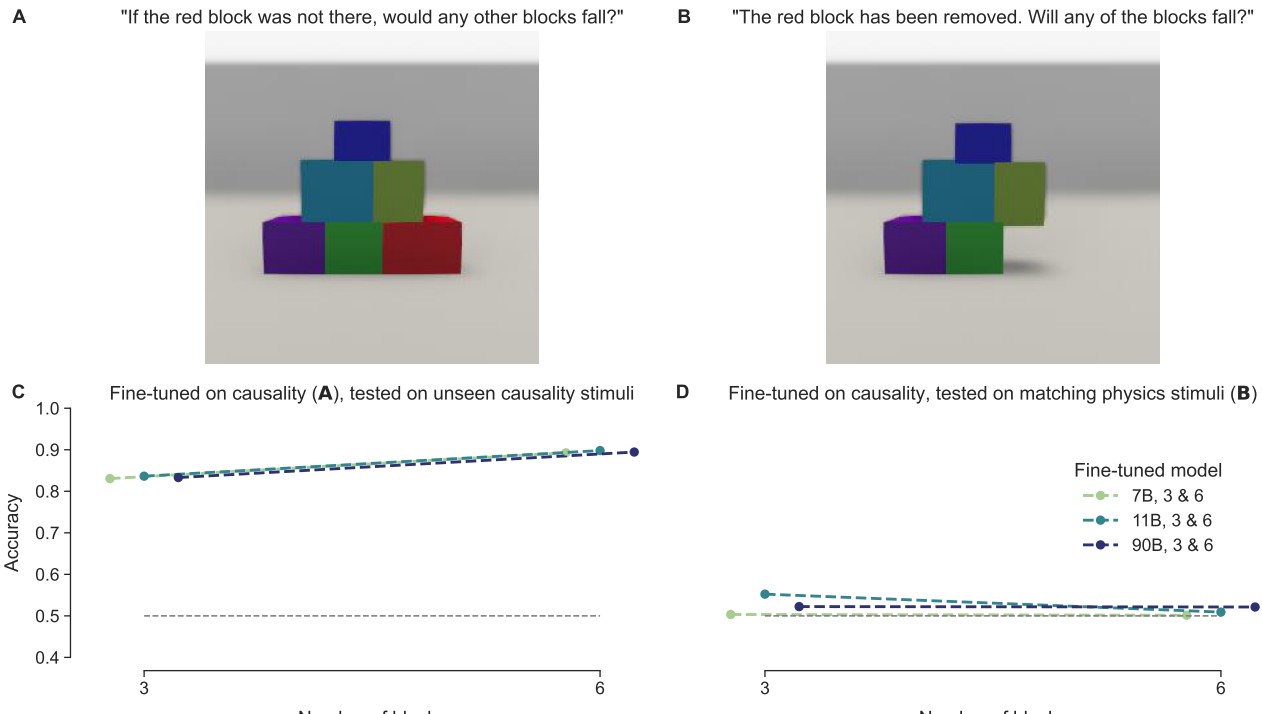

*Figure 4.* **A:** Models are fine-tuned on the counterfactual reasoning task with pyramids of 3 & 6 blocks. **B:** Models are given the corresponding images to their training data with the red block removed, and must judge whether it is stable. **C:** Models can generalize to unseen pyramids on the causal reasoning task. **D:** Models cannot generalize to judging the factual stability of the pyramids they have been trained on, only now without the red block.

We also fine-tune joint models on combined halves of 5,000 data points from each domain. We again find that these models do not perform well on all Lerer towers (see *Joint* rows in Fig. 2). Indeed, there appears to be a trade-off where models that perform well on 2–4 block towers in *Cubeworld* perform poorly on 2 block towers from the Lerer dataset.

### 3.4. Generalizing to a new task

We find that no model fine-tuned on a single cognitive domain performs well on the other cognitive domain (see Fig. 2 and Figs. 14–16 in Appendix G). Models were fine-tuned on intuitive physics towers or causal reasoning pyramids from *Cubeworld*. To test how well models generalize to another task in another cognitive domain, we evaluate them on the task they were not fine-tuned on.

Reasoning about tower stability is a prerequisite for counterfactual judgments on tower stability. This is especially obvious for the 3 block tower pyramids, where computing the counterfactual requires a tower stability judgment on a two block tower. Therefore, we would expect an improvement in causal reasoning to carry with it improvement on intuitive physics as well.

However, models fine-tuned on a mixture of data from both tasks can achieve good performance in both domains, with only slight performance decrements in either domain. This confirms that the models have the capacity of solving both tasks at the same time. Still, models fine-tuned on a single cognitive domain are unable to generalize to the other domain.

To establish whether these failures to generalize to other tasks are due to small differences between tasks, or if the models struggle with learning intuitive theories through task-specific fine-tuning, we added another dataset where differences between the tasks are kept as minimal as possible. We generate paired images of pyramids, in which the causal reasoning image contains a red block which is removed to generate the intuitive physics image (see Fig. 4).

In principle, being able to reason about the counterfactual stability of a pyramid ought to predispose models to reason about the factual stability of pyramids. Thus, we expected a transfer from causal reasoning to intuitive physics, especially since we test models using the corresponding images from the pairs they were fine-tuned on. Furthermore, we explicitly tell the models that the red block has been removed. Nevertheless, we do not find evidence of this transfer, suggesting that task-specific fine-tuning does not lead to models learning intuitive theories. Instead, they appear to be learning task-specific superficial shortcuts that do not generalize (Geirhos et al., 2020a; Ilyas et al., 2019).

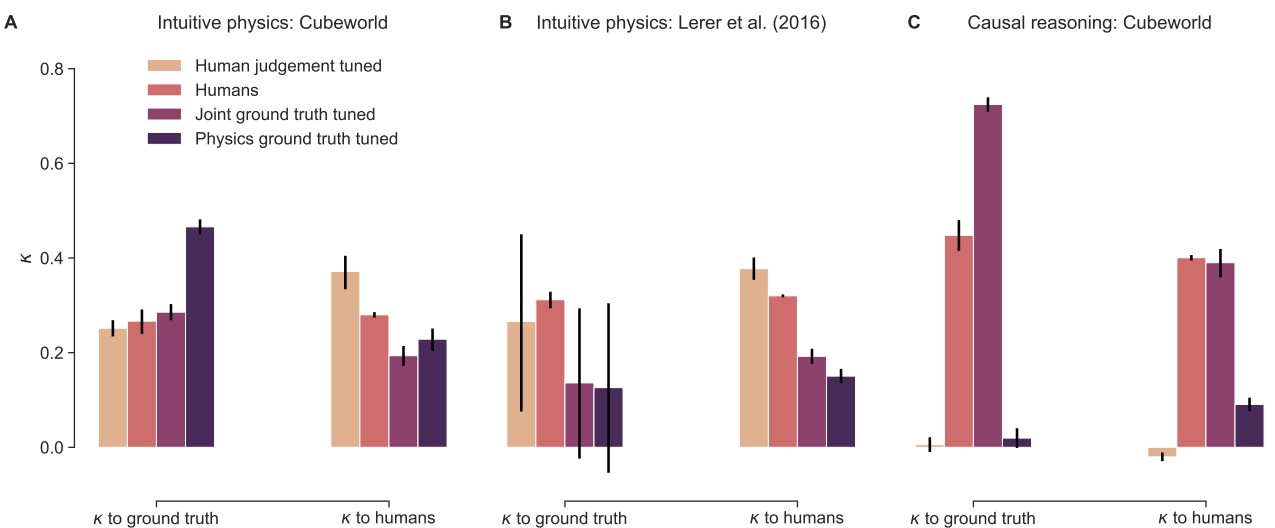

*Figure 5.* Error consistency to ground truth and human raters on three evaluation datasets (A–C) for the 11B model separately fine-tuned on three different datasets: (1) human judgments on intuitive physics, (2) the ground truth on intuitive physics, (3) the ground truth on causal reasoning and intuitive physics. Human error consistency is provided as a comparison. **A:** Results for the *Cubeworld* intuitive physics evaluation dataset. **B:** Results for the naturalistic dataset (Lerer et al., 2016). **C:** Results for the *Cubeworld* causal reasoning evaluation dataset.

### 3.5. Alignment with human judgments

We see that fine-tuning on the ground truth leads to some alignment with human judgments on the fine-tuning task. However, this does not transfer well to human judgments on the same task with other visual characteristics, and not at all to human judgments on another cognitive domain.

To analyze the alignment of model judgments with human judgments, we use boostrapped Cohen's $\kappa$ arithmetic means (Geirhos et al., 2020b), a single behavioral score that measures the agreement between two observers from their responses (see Appendix D).

The 11B model fine-tuned on the ground truth intuitive physics 2–4 block towers have a mean $\kappa$ of 0.23 with humans on the same task, but only 0.15 on the Lerer task, and 0.09 on the causal reasoning pyramids (see Fig. 5). In contrast, the 11B model fine-tuned on the ground truth causal reasoning 3 & 6 block pyramids has a mean $\kappa$ of 0.4 with humans on the same task, but only $-0.02$ on the Lerer task, and $-0.08$ on the intuitive physics block towers.

### 3.6. Fine-tuning on human data

We find that fine-tuning the model on human judgments makes model behavior more similar to human behavior. We collected single human responses for each of the intuitive physics 2–4 block towers in the fine-tuning data set (see section 2.5), allowing us to fine-tune models on the human responses instead of the ground truth. The correlation between the human fine-tuning data and the ground truth was 0.27, with an overlap in labels of 63.5%.

We find that fine-tuning on human responses leads to a considerable performance improvement on judging the ground truth stability of intuitive physics block towers compared to the base model. Even though the model is fine-tuned on human judgments on towers of size 2–4, it learns to predict the ground truth stability of bigger towers as well. We see the same pattern emerge as before, where model accuracy decreases as the number of blocks increases, albeit with overall slightly lower accuracies compared to the model fine-tuned on the ground truth.

Furthermore, fine-tuning the model on human judgments increases the mean $\kappa$ with human judgments on the same data set to 0.37 (see Fig. 5A). Additionally, it leads to a higher mean $\kappa$ of 0.37 with human judgments on the same task with different visual characteristics and a better transfer on the ground truth performance (see Fig. 5B). It however does not transfer to the causal reasoning domain (see Fig. 5C). Here, the mean $\kappa$ to humans is $-0.02$.

## 4. Discussion

Previous work has shown that pre-trained VLMs struggle with several aspects of visual cognition, particularly in causal reasoning and intuitive physics (Schulze Buschoff et al., 2025). We find that fine-tuning on intuitive physics and causal reasoning tasks can improve the performance of VLMs in these cognitive domains, and that it improves alignment with human judgments. However, there is also evidence that fine-tuned VLMs' physical and causal reasoning is brittle. On the naturalistic intuitive physics data from

Lerer et al. (2016), models fine-tuned on *Cubeworld* data in the same domain and task perform below chance level for some tower sizes.

Similarly, neither models fine-tuned on intuitive physics nor models fine-tuned on causal reasoning successfully generalize to the other cognitive domain. This result is particularly notable for the models fine-tuned on causal reasoning, as the ability to make physical stability judgments is a necessary precursor capability for judging the counterfactual stability of a tower. Models' inability to generalize to another cognitive domain is not due to them being limited in parameters or potential ability — models fine-tuned on a mixture of intuitive physics and causal reasoning data performed well in both domains. It is important to note that we primarily showcase the limits of models fine-tuned on a specific task. While we cannot evaluate how the joint models would generalise to a third cognitive task in *Cubeworld*, it is possible that fine-tuning models on broader distributions of tasks could lead to more robust improvements. Indeed, models fine-tuned on data from both tasks also somewhat generalize to the realistic block towers from Lerer et al. (2016), suggesting that data diversity is beneficial for generalization performance. These models also show higher agreement with human judgments on both the artificial and realistic datasets.

One account for these results is that fine-tuning does not reduce the effect of the so-called *binding problem* (Campbell et al., 2024; Frankland et al., 2021). In human visual cognition, participants placed under significant cognitive load by having to process multiple multi-feature objects very quickly make more mistakes than usual. In our tasks, models had to process multiple blocks simultaneously, judging their colours and relative positions. While fine-tuning improves performance on specific tasks, perhaps facilitating a better division of labour between specific feature detectors, these strategies are brittle and appear to fail on novel domains. An alternative account is that supervised fine-tuning leads to data memorization, whereas a reinforcement-learning-based post-training method would better facilitate generalization (Chu et al., 2025). We leave exploring these hypotheses to future work.

We present a first investigation on the limitations of fine-tuning for visual cognition. There are several avenues for future research to improve our understanding of fine-tuning and how well fine-tuned models can generalize:

First, it is possible that robust generalization from the fine-tuning domain to another can only emerge with even larger models. We studied the effects of fine-tuning on models up to 90 billion parameters, which are relatively small compared to some closed-source alternatives. Future work should therefore explore fine-tuning to improve even larger models. Second, alternative fine-tuning procedures may

improve outcomes. While we do not find evidence of overfitting *per se*, it is possible that the models have overfitted in a more general, task-level sense. The models may have been sensitive to non-robust predictive features of the fine-tuning data in a particular domain that led to good performance there but not on new domains or with naturalistic data (Ilyas et al., 2019; Geirhos et al., 2020a). Parametrizing the QLoRA procedure with different $r$ and $\alpha$ may improve generalization performance by modulating knowledge distillation and the relative effects of model weights and adapter weights. Similarly, introducing greater variance into the fine-tuning datasets, fine-tuning on broader distributions of tasks, and fine-tuning on larger volumes of data might improve model performance. Finally, the visual and cognitive demands of these tasks are entangled. Models require visual abilities to detect and distinguish blocks, appraise distances, and judge three-dimensional volumes from two-dimensional images. They also need an understanding of gravity, mass, inertia, and friction. Future work should explore whether providing information about these properties in symbolic or schematic form can lead to improved performance and better generalization in these tasks.

Our findings underscore the limits of task-specific parameter-efficient fine-tuning in achieving robust generalization in vision-language models. PEFT noticeably improves performance on tasks closely similar to the fine-tuning data, enabling generalization not just to new data sampled from the fine-tuning distribution but also to, for example, taller and shorter towers and pyramids—an out-of-distribution problem. Moreover, PEFT can align models more closely with human behavior in these contexts. However, task-specific fine-tuning does not lead to the broad, flexible reasoning abilities that characterize human cognition. Models fine-tuned on one cognitive task (e.g., intuitive physics) fail to generalize to another (e.g., causal reasoning), despite clear conceptual overlap, and models struggle to reliably extrapolate their knowledge to real-world images with different visual properties.

Together, these results suggest that current approaches to fine-tuning are limited in the ways that they can improve models, and remain insufficient for developing models that reason about the physical and causal structure of the world in a way that mirrors human cognition. Achieving this level of generalization may require different training and fine-tuning paradigms that go beyond parameter-efficient adaptation.

## Acknowledgements

We thank Marcel Binz, Can Demircan, and Julian Coda-Forno for helpful discussions and feedback on the manuscript.

## Impact Statement

This paper presents work whose goal is to advance the field of Machine Learning. There are many potential societal consequences of our work, none of which we feel must be specifically highlighted here.

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

# A. Data examples

## A.1. Intuitive physics

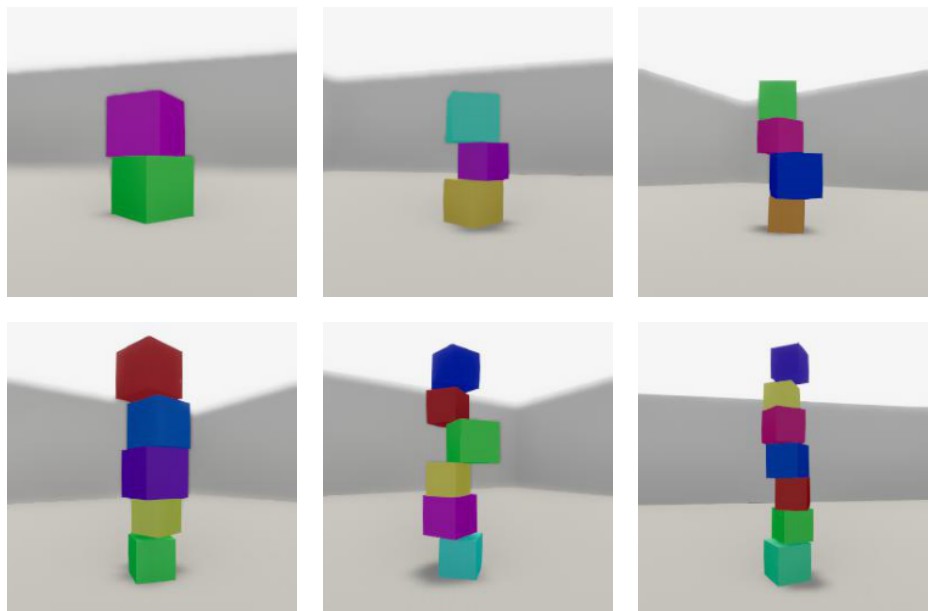

*Figure 6.* Stable examples from the *Cubeworld* intuitive physics block tower evaluation set.

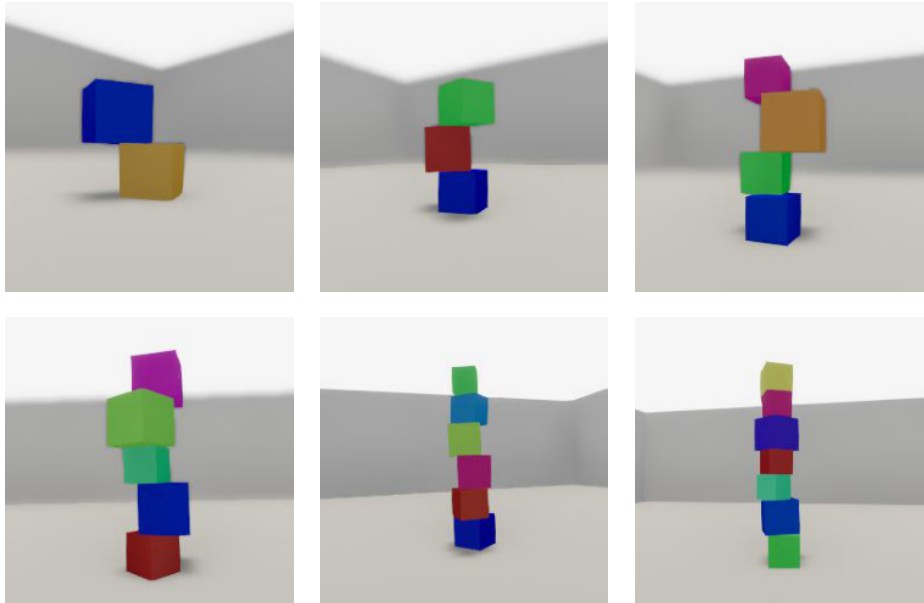

*Figure 7.* Unstable examples from the *Cubeworld* intuitive physics block tower evaluation set.

## A.2. Causal reasoning

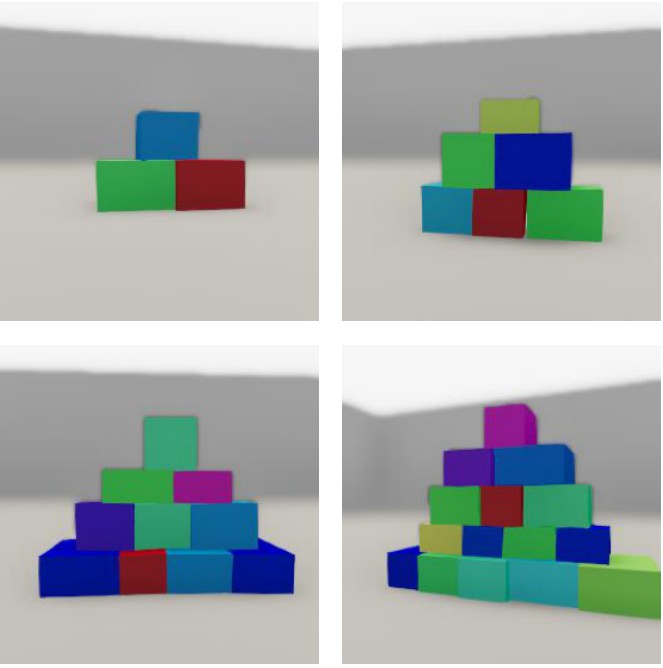

*Figure 8.* Stable examples from the *Cubeworld* causal reasoning pyramid evaluation set.

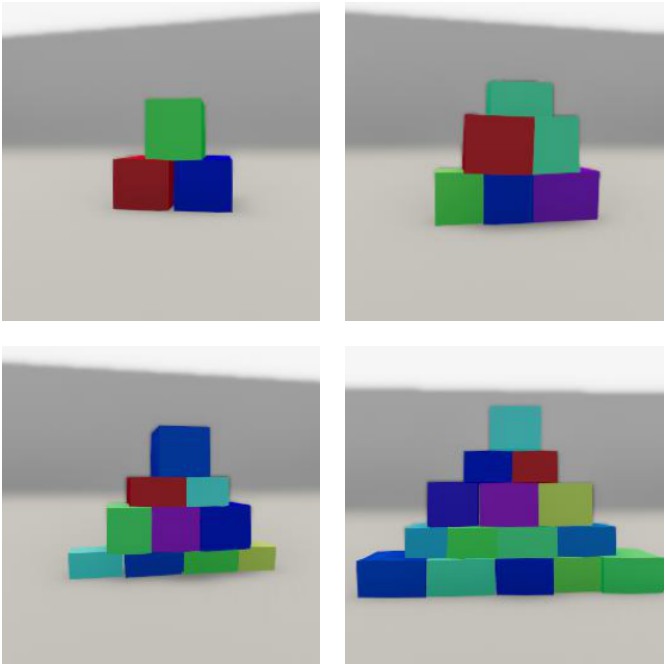

*Figure 9.* Unstable examples from the *Cubeworld* causal reasoning pyramid evaluation set.

## A.3. Lerer

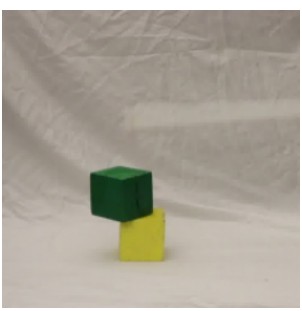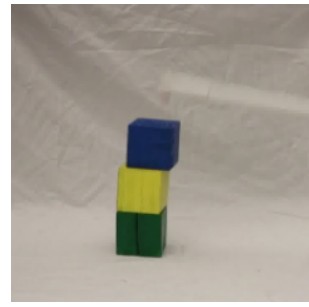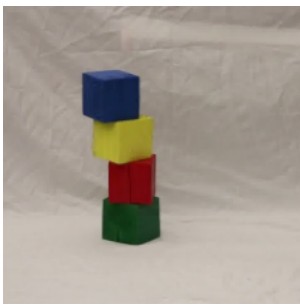

*Figure 10.* Examples from the Lerer et al. (2016) evaluation set.

# B. Human experiment information and demographics

**Intuitive physics: human fine-tuning data**   Participants received a base pay of 1.5\$ and an additional bonus of 0.01\$ for each correct answer, bringing the maximum payout to 2.5\$. Completing the experiment took participants 09:54 minutes on average. All participants were native English speakers from the UK and the US with a mean age of 30.34 and a split of 55 females to 57 males.

**Intuitive physics: human evaluation data**   Participants received a base pay of 1.5\$ for their participation and an additional bonus of 0.01\$ for each correct answer, bringing the maximum payout to 2.7\$. Completing the experiment took participants 11:37 minutes on average. All participants were native english speakers from the UK and the US and had a mean age of 30.46 and a split of 50 females to 50 males.

**Causal reasoning: human evaluation data**   Participants received a base pay of 1\$ for their participation and an additional bonus of 0.01\$ for each correct answer, bringing the maximum payout to 1.8\$. Completing the experiment took participants 09:07 minutes on average. All participants were native English speakers from the UK and the US with a mean age of 31.32 and a split of 50 females to 50 males.

# C. Analysis Tools

We analyze all data using Python 3.12.7 using pandas 2.2.2 (pandas Development Team, 2020), seaborn 0.13.2 (Waskom, 2021), matplotlib 3.9.2 (Hunter, 2007), and statsmodels 0.14.2 (Seabold & Perktold, 2010).

# D. Analysis Methods

We use Cohen's $k$ to analyze the alignment of models to human judgments and the ground truth in Figure 5. Cohen's $k$ is define as:

$$\kappa_{i,j} = \frac{c_{\text{obs}_{i,j}} - c_{\text{exp}_{i,j}}}{1 - c_{\text{exp}_{i,j}}}$$

where $c_{\text{obs}}$ is the observed error overlap defined as $c_{\text{obs}_{i,j}} = \frac{e_{i,j}}{n}$ with $e_{i,j}$ as the number of equal responses and $c_{\text{exp}} = p_i p_j + (1 - p_i)(1 - p_j)$, the expected overlap that two observers $i$ and $j$ with accuracies $p_i$ and $p_j$ will have by chance.

To arrive at a single mean $\kappa$ between models and humans, we calculate $\kappa$ for each combination of models and humans and take the mean over the $\kappa$ values. We produce bootstrapped 95% confidence intervals by computing the central 95 percentiles over 10,000 random subsamples of the $\kappa$ distribution. For comparisons to the ground truth, we use the central 95 percentiles of the distribution over mean $\kappa$ for 10,000 random subsamples of the item level judgments. We note that models fine-tuned on human judgments have a higher $\kappa$ between the model and humans than between the humans themselves (see Figure 5). This is likely because the latter calculation has many more degrees of freedom than the former since every human is compared to every other, whereas each human is compared only to one model at a time.

# E. Performance improvement over time

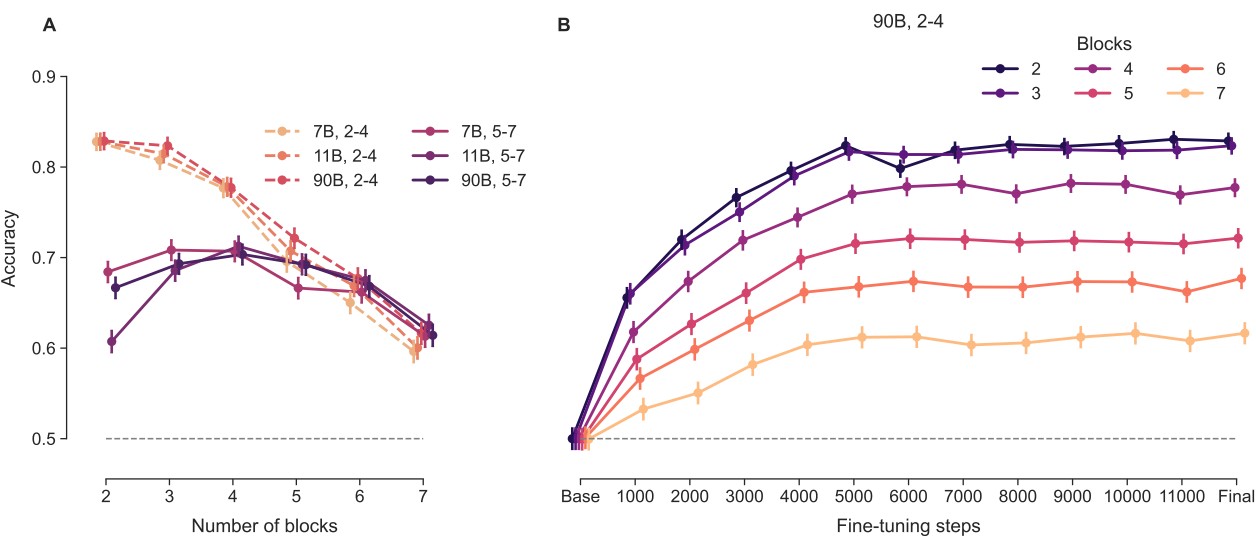

*Figure 11.* **A:** Models evaluated on intuitive physics tower stacks that either have the same number of blocks as the fine-tuning data or different numbers of blocks compared to the fine-tuning data. Note: bars are jittered slightly for better readability. **B:** Performance of the 90B model fine-tuned on 2–4 blocks on the test set for all number of blocks over the process of fine-tuning. The model performs best on towers it is fine-tuned on but it can generalize to bigger towers somewhat. Generalization decreases as the block tower size moves away from the fine-tuning distribution. Both subplots show the proportion of correct answers (accuracy) with Wilson score intervals as error bars.

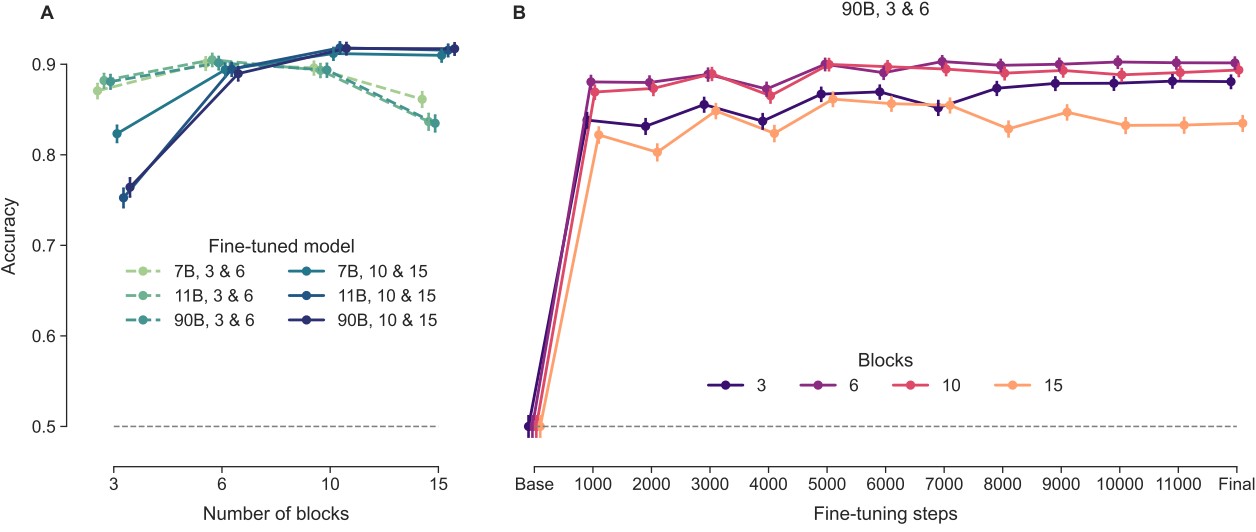

*Figure 12.* **A:** Models evaluated on causal reasoning pyramids that either have the same number of blocks as the fine-tuning data or different numbers of blocks compared to the fine-tuning data. Models performance is roughly the same for in-distribution and out-of-distribution pyramid sizes. Note: bars are jittered slightly for better readability. **B:** Performance of the 90B model fine-tuned on 3 & 6 block pyramids on the test set for all number of blocks over the process of fine-tuning. The model performs well on the pyramids it is fine-tuned on and can generalize to bigger towers. Both subplots show the proportion of correct answers (accuracy) with Wilson score intervals as error bars.

# F. Generalization to Lerer

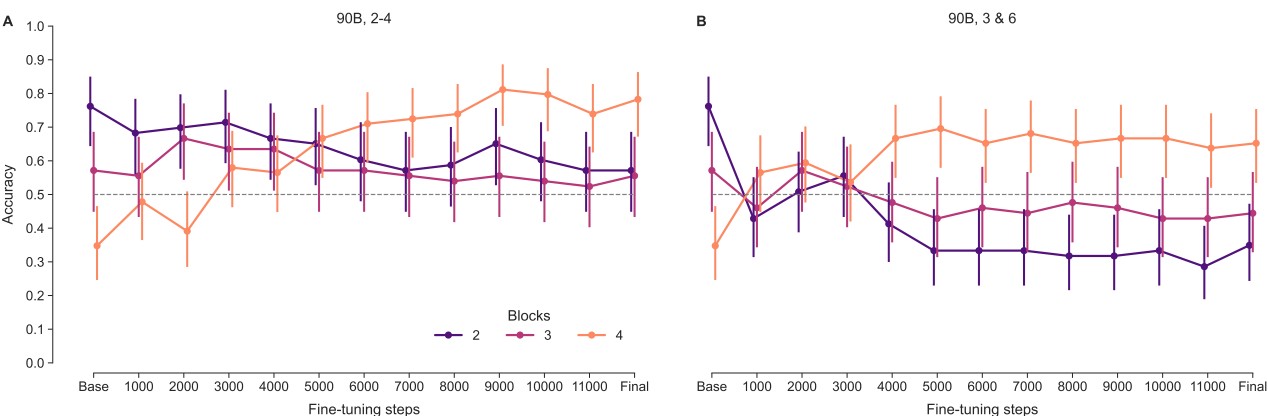

*Figure 13.* **A:** 90B model fine-tuned on intuitive physics towers with 2–4 blocks but evaluated on the Lerer stimuli showing real images of 2–4 block towers. **B:** 90B model fine-tuned on 3 & 6 block pyramids, evaluated on the Lerer stimuli, which have a different visual quality and are in another cognitive domain.

# G. Generalization to new domains over time

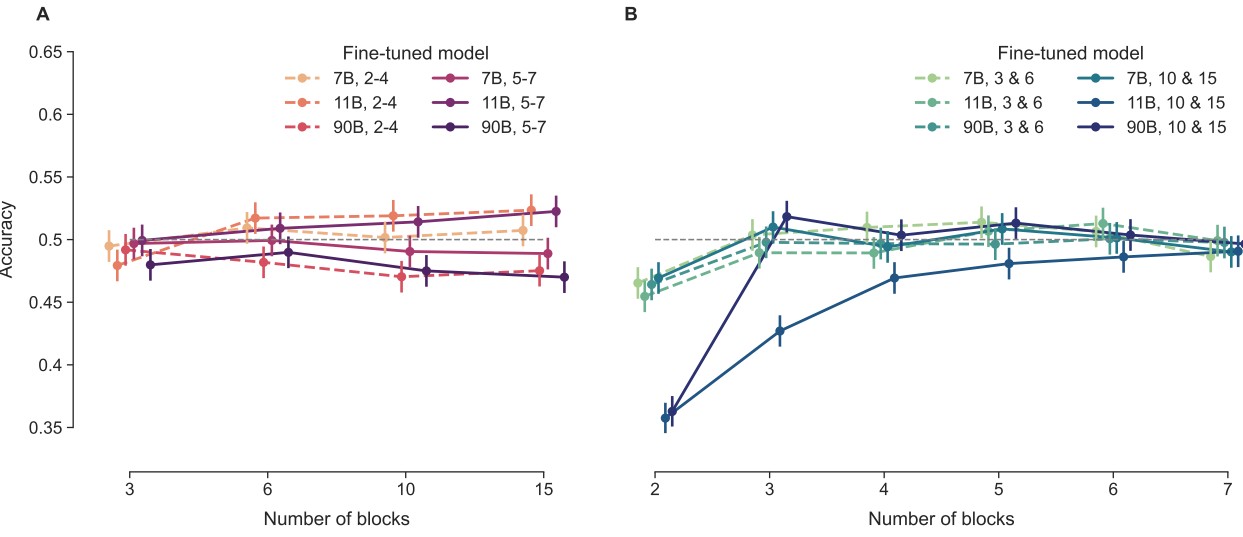

*Figure 14.* **A:** Models fine-tuned on intuitive physics towers but evaluated on causal reasoning pyramids. **B:** Models fine-tuned on causal reasoning pyramids but evaluated on intuitive physics towers. Models do not generalize to the other domain, even though it shares the same visual features.

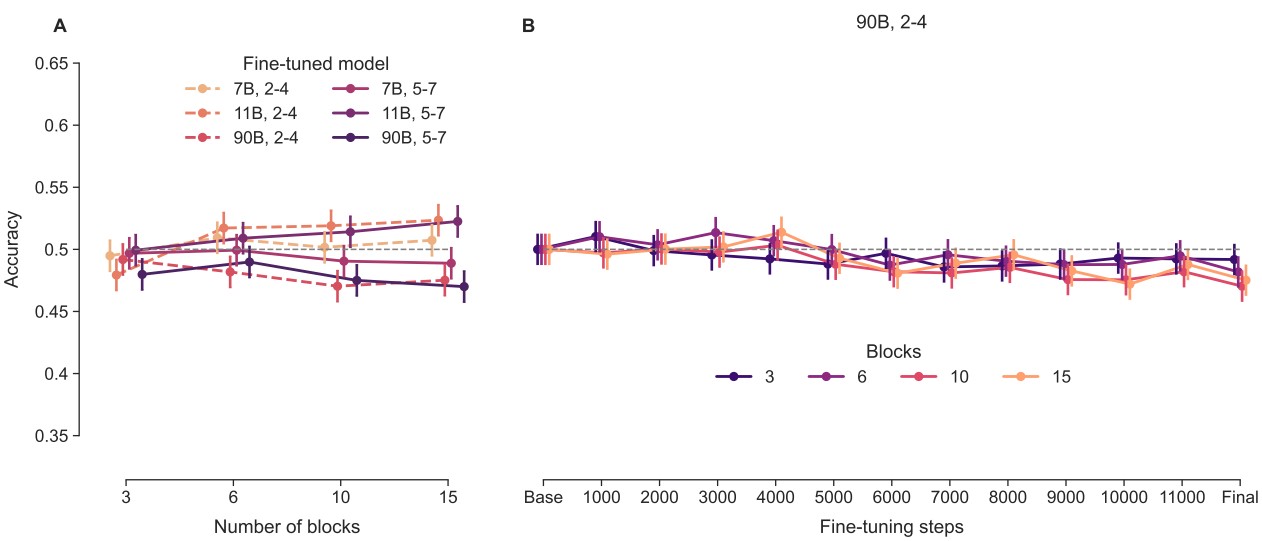

*Figure 15.* **A:** Models fine-tuned on intuitive physics towers but evaluated on causal reasoning pyramids. Models again do not generalize to this other task, even though it shares the same visual features. **B:** Performance of the 90B model fine-tuned on 2–4 block towers, tested on causal reasoning pyramids. The model does not generalize to any tower size. Both subplots show the proportion of correct answers (accuracy) with Wilson score intervals as error bars.

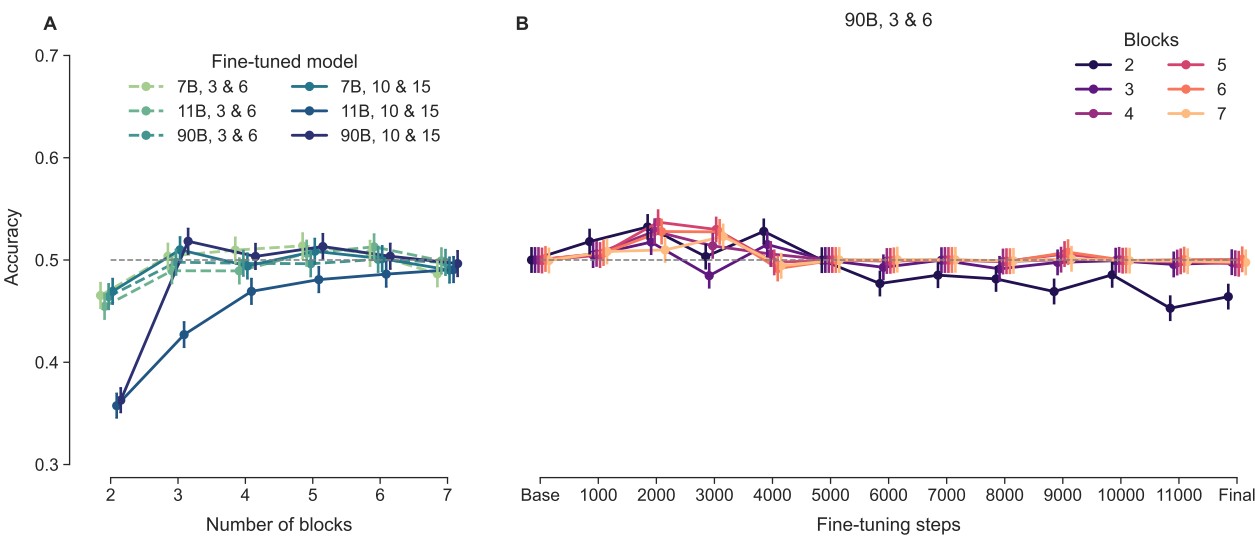

*Figure 16.* **A:** Models fine-tuned on causal pyramids but evaluated on intuitive physics tower stacks. Models do not generalize to this other task, even though it shares the same visual features and can be seen as a pre-requisite for the fine-tuning task. This is especially true for the fine-tuning on 3 block tower pyramids, where computing the counterfactual question requires solving the binary tower stability of a two block tower. **B:** Performance of the 90B model fine-tuned on 3 & 6 block pyramids, tested on intuitive physics block towers. The model does not generalize to any tower size. Both subplots show the proportion of correct answers (accuracy) with Wilson score intervals as error bars.

