# OpenReview forum: "Testing the Limits of Fine-Tuning for Improving Visual Cognition in Vision Language Models"
_ICML.cc/2025/Conference — ICML 2025 poster_

### Official Review · Reviewer_3bXm · 2025-03-13

**Overall Recommendation:** 4

**Summary:**

This work investigates the extent to which task-specific fine-tuning can help VLMs to overcome limitations in two psychologically inspired domains, with a special emphasis on the extent to which benefits of fine-tuning generalize between tasks and across variations within a task.

**Claims And Evidence:**

The experiments very clearly demonstrate that task-specific fine-tuning leads to brittle improvements, in the sense that these improvements do not generalize between tasks, and they do not even clearly generalize to slightly different versions of the same task. I have a few comments about the interpretation of these results. First, the experiments are described as establishing the limits of fine-tuning in general, but it seems more accurate to say that they establish the limits of *task-specific* fine-tuning. It would be good to clearly state that the experiments do not necessarily rule out more robust gains from fine-tuning across a wider distribution of tasks (and they even seem to suggest that this might work, given the relatively improved generalization in the model fine-tuned on both tasks). Second, it seems that the fine-tuned models actually outperform humans within the specific domain that they are fine-tuned in (although this is difficult to tell because the data are not presented in the same plot). In other words, the results clearly demonstrate that fine-tuning yields only very brittle benefits, which does not seem very human-like, but it seems worth mentioning that the performance is actually quite strong relative to humans within these domains.

**Essential References Not Discussed:**

N/A

**Experimental Designs Or Analyses:**

All experiment designs and analyses appear reasonable.

**Methods And Evaluation Criteria:**

The methods and evaluation criteria are all sensible. In Figure 2, it would be helpful to include additional rows illustrating 1) zero-shot performance of the base model, 2) human performance, 3) zero-shot performance for a state-of-the-art model (e.g. GPT-4o or Claude). This will help to assess not only how fine-tuning generalizes between tasks, but also how the performance of the fine-tuned models compares in absolute terms with humans and other models.

**Other Comments Or Suggestions:**

In Figure 4, why is the model-to-human alignment apparently higher than the human-to-human alignment in some cases (i.e. the bars labelled 'humans')?

**Other Strengths And Weaknesses:**

N/A

**Questions For Authors:**

N/A

**Relation To Broader Scientific Literature:**

The discussion of the relationship to the broader literature is generally strong. There are two points where it may be expanded. First, some recent work [1] has found that failures of VLMs can be related to the classic 'binding problem' in cognitive science, in the sense that the failures are similar to those observed for human participants under time pressure. This is also related to theoretical work [2] which suggests that sequential processing (i.e. inference-time compute) is needed to overcome these binding errors. One prediction of this perspective is that, so long as the model is required to generate a response in a single feedforward pass, fine-tuning will only lead to very task-specific benefits, whereas inference-time compute will be needed for more generalizable benefits. The results in this paper seem consistent with that line of reasoning, which may be interesting to discuss.

Additionally, it may be interesting to discuss whether approaches like [3] may be lead to more generalizable benefits, i.e. via fine-tuning on  a broader distribution of tasks.

[1] Campbell, D., Rane, S., Giallanza, T., De Sabbata, C. N., Ghods, K., Joshi, A., ... & Webb, T. (2024). Understanding the limits of vision language models through the lens of the binding problem. Advances in Neural Information Processing Systems, 37, 113436-113460.

[2] Frankland, S. M., Webb, T. W., Lewis, R. L., & Cohen, J. D. (2025). No Coincidence, George: Processing Limits in Cognitive Function Reflect the Curse of Generalization.

[3] Binz, M., Akata, E., Bethge, M., Brändle, F., Callaway, F., Coda-Forno, J., ... & Schulz, E. (2024). Centaur: a foundation model of human cognition. arXiv preprint arXiv:2410.20268.

**Theoretical Claims:**

N/A

---

> ### Author Rebuttal · Authors · 2025-03-30
>
> Dear reviewer 3bXm, thank you very much for your comments. We are happy to hear that our experiments “very clearly demonstrate that task-specific fine-tuning leads to brittle improvements”, this was the main point we were trying to convey. We also appreciate your assessment that our “methods and evaluation criteria are all sensible” and that you found our discussion of related work “generally strong”. In the following we discuss the specific concerns that you raised and how we have sought to remedy them. We think addressing them has significantly strengthened our paper, particularly with respect to improving the clarity and including a GPT baseline.
> \
> \
> *The experiments are described as establishing the limits of fine-tuning in general, but it seems more accurate to say that they establish the limits of task-specific fine-tuning.*
> \
> \
> Thank you for raising this point. We agree that our experiments first and foremost establish the limits of fine-tuning on specific tasks, namely, reasoning about the stability of solid block towers. However, we would like to emphasise the long history of using such tasks for investigating visual cognition in humans. We think that they are sufficiently representative for us to make some more general claims about visual cognition in vision language models. Our results present some evidence that fine-tuning models on one cognitive task does not lead to generalization on a related but distinct cognitive task. In contrast to this, human learning is characterised by a robust ability to generalise between distinct but related tasks.
> \
> \
> You are correct to point out that models trained on both tasks do generalize slightly better than models trained on only one (Fig. 2). However, with our current datasets, we cannot evaluate how these models would generalise to a third cognitive task in cubeworld, and we suspect they would generalize poorly, given our current results.
> \
> \
> *Second, it seems that the fine-tuned models actually outperform humans within the specific domain that they are fine-tuned in (although this is difficult to tell because the data are not presented in the same plot). In other words, the results clearly demonstrate that fine-tuning yields only very brittle benefits, which does not seem very human-like, but it seems worth mentioning that the performance is actually quite strong relative to humans within these domains.*
> \
> \
> Thank you for these comments. First, we have addressed the problem of model-human comparison by including a human baseline in Figure 2, and all heatmaps in the appendices. Second, we have emphasised in the discussion that fine-tuned models outperform humans on the domains they have been directly trained on, while being outperformed by them on out-of-distribution domains.
> \
> \
> *In Figure 2, it would be helpful to include additional rows illustrating 1) zero-shot performance of the base model, 2) human performance, 3) zero-shot performance for a state-of-the-art model (e.g. GPT-4o or Claude).*
> \
> \
> Thank you for this suggestion. We agree that this makes it easier to interpret the results. We have therefore added rows showing the zero-shot performance of the base model, human performance and zero-shot performance for GPT-4o. You can see the updated figure here: [Figure 2](http://postimg.cc/8JwwDwYv) (please note this shows Llama-3.2-11B, as other reviewers highlighted the need for bigger models from other families).
> \
> \
> *Relation to broader scientific literature could be expanded.*
> \
> \
> Thank you for the very relevant pointers. We have updated our related works section according to the feedback by you and by reviewers fjkm and uVR9. We were aware of [2] but had not made the connection to the difficulties that models have with generalization. This is a very interesting direction and we thank you for bringing it to our attention. We have also added [3] to better motivate why fine-tuning on human choices is of interest and might lead to more robust generalization.
> \
> \
> *In Figure 4, why is the model-to-human alignment apparently higher than the human-to-human alignment in some cases (i.e. the bars labelled 'humans')?*
> \
> \
> This is an intriguing outcome from our initial results. It indicates that the average agreement between the model and each human rater is higher than the average agreement between every pair of human raters. We speculate that this arises from the fact that there is a sizable variance in human ratings with heavy tails; the fine-tuned model appears to accurately capture the average human rating, resulting in a higher average agreement than the human case, where average agreement is pulled down by raters with different ratings due to the way Cohen’s kappa is computed, since the observed agreement will be low relative to the expected agreement. We should note that our plots have changed slightly, since we now average over three fine-tuning seeds (see the updated [Figure 4](http://postimg.cc/47qjMqfW) with the 7B Qwen model).

---

> > ### Comment · Reviewer_3bXm · 2025-04-03
> >
> > Thank you to the authors for these replies. I appreciate the updates to Figure 2, which I think will make it easier to interpret the results. I still think that the paper would be improved if it were clearly stated that the experiments primarily establish the limits of *task-specific* fine-tuning. I very much agree that the tasks are 'sufficiently representative to make some more general claims about visual cognition in vision language models'. The tasks are reasonable for establishing the visual cognitive abilities of these models. But this doesn't imply that fine-tuning on only these tasks can comprehensively establish the limits of fine-tuning, whether task-specific or over a more general distribution of tasks. It seems at least possible that fine-tuning over a broader distribution of tasks could lead to more robust improvements. I think this is primarily a matter of clearer framing, and not a fundamental criticism of the work, which I think makes a useful contribution.

---

> > > ### Author Response · Authors · 2025-04-04
> > >
> > > Dear reviewer 3bXm, we again want to thank you for your time and for actively taking part in the discussion process. We are happy to hear that our changes improved the interpretability of the results. We understand your remaining concerns about task-specificity – we agree that our results primarily showcase the limits of fine-tuning on a specific task, even if it is representative for a given cognitive domain. We made the following changes to the introduction to highlight this more clearly (changes marked in bold).
> > >
> > >
> > > - Abstract line 23: *However, we find that **task-specific** fine-tuning does not contribute to robust human-like generalization to data with other visual characteristics or to tasks in other cognitive domains.*
> > > - Introduction line 84 left: *In this paper, we explore whether fine-tuning VLMs **on single tasks** can improve their performance on intuitive physics and causal reasoning tasks in the visual domain, as well as steer them towards more human-aligned outputs.*
> > > - Introduction line 93 left: ​​*Therefore, we seek to evaluate whether **task-specific** fine-tuning not only improves performance on visual cognition tasks sampled from an identical distribution, but also whether it produces models that can generalize to new, but related, tasks in new domains.*
> > > - Introduction line 102 left: *Our results allow us to appraise the limits of **task-specific** fine-tuning for building performant, human-like machine learning models that can generalize beyond the kinds of data on which they have been trained. Across a range of datasets and models, we do not find evidence that fine-tuning alone can achieve all these objectives.*
> > > - Introduction line 101 right: *In this work, we fine-tune VLMs on **single tasks** from two cognitive domains, intuitive physics and causal reasoning, using tasks designed in a virtual environment we call Cubeworld, in which block stacks are constructed from colored cubes and are subject to realistic physical forces.*
> > >
> > >
> > >
> > > We have also added further clarification about this constraint in the limitations section:
> > >
> > > - Discussion line 408 left: *\[...\] models fine-tuned on a mixture of intuitive physics and causal reasoning data performed well in both domains. It is important to note that we primarily showcase the limits of models fine-tuned on a specific task. **While we cannot evaluate how the joint models would generalise to a third cognitive task in Cubeworld, it is possible that fine-tuning models on broader distributions of tasks could lead to more robust improvements.***
> > > - Discussion line 391 right: *Similarly, introducing greater variance into the fine-tuning datasets, **fine-tuning on broader distributions of tasks,** as well as fine-tuning on larger volumes of data might improve model performance.*
> > > - Discussion line 403 right: *Our findings underscore the limits of **task-specific** fine-tuning in achieving robust generalization in vision-language models.*
> > > - Discussion line 412 right: *However, **task-specific** fine-tuning does not lead to the broad, flexible reasoning abilities that characterize human cognition*
> > >
> > >
> > >
> > > We sincerely appreciate your thoughtful assessment of our work and are glad that you see it as a useful contribution. We hope that our clarifications and improvements regarding the framing of our work have successfully addressed your concerns and that, in light of these revisions, you may now see it as a clear accept.

---

### Official Review · Reviewer_uVR9 · 2025-03-14

**Overall Recommendation:** 2

**Summary:**

This paper investigates whether fine-tuning models on intuitive physics and causal reasoning improves performance within these specific domains. The authors conclude that such fine-tuning does not enhance performance on other visual characteristics or tasks in different cognitive domains.

**Claims And Evidence:**

The submission is supported by some evidence. As demonstrated in Figure 2, fine-grained control variable experiments were conducted across various settings, including different fine-tuning datasets and evaluation sets. However, the experiments only utilize Qwen2-VL models with 2B and 7B parameters. I believe that testing a broader range of model architectures, such as LLaVA, and including larger models, like those with 30B or 70B parameters, would strengthen the conclusions.

**Essential References Not Discussed:**

This observation is linked to the discussion on the generalization of reasoning in large language models (LLMs) and vision-language models (VLMs). In the context of LLMs, it relates to references [1][2][3], which explore whether these models, when fine-tuned on specific tasks, can generalize to in-domain/out-of-domain or simpler/more complex tasks. I believe these discussions are also relevant to this paper's ideas, as the understanding and intelligence of VLMs are fundamentally derived from LLMs. Consequently, grasping the generalization capabilities of LLMs is essential for comprehending the generalization of VLMs. For vision-language models, this is connected to reference [4].

[1] Faith and Fate: Limits of Transformers on Compositionality. Nouha Dziri et al.
[2] Algorithms Can Transformers Learn? A Study in Length Generalization. Hattie Zhou et al.
[3] Math for AI: On the Generalization of Learning Mathematical Problem Solving. Ruochen Zhou et al.
[4] In-Context Compositional Generalization for Large Vision-Language Models. Chuanhao Li et al.

**Experimental Designs Or Analyses:**

I have reviewed the experimental designs presented in Figures 2, 3, and 4, and they appear to be fundamentally sound and valid. However, the key points in Figures 3 and 4 are not clearly conveyed, making them difficult to understand. I recommend that the authors add annotations to these figures to clarify the key points.

**Methods And Evaluation Criteria:**

It appears that all experiments are conducted on the cubes dataset; I suggest that incorporating other types of physical understanding tasks could also help verify the findings.

**Other Comments Or Suggestions:**

No.

**Other Strengths And Weaknesses:**

This work addresses the generalization problem of vision-language models (VLMs), which is an intriguing topic. However, the experiments do not provide sufficient analysis on the reasons behind the failures in generalization, which limits the insights offered by the study. Furthermore, I find that the paper is poorly written and lacks organization. I recommend that the authors reorganize the experimental sections and clarify the conclusions. For example, Figures 3 and 4 are challenging to interpret, and the descriptions associated with these figures could be placed in separate sections for enhanced clarity. Additionally, restructuring the discussion section could help present the key points more effectively.

**Questions For Authors:**

1. What are the underlying reasons for the failures in generalization? Could the authors provide a more detailed analysis of these factors?
2. Will larger models exhibit different patterns regarding this phenomenon? Like 13B/30B/70B models?

**Relation To Broader Scientific Literature:**

See the below session.

**Theoretical Claims:**

There are no equation in this paper.

---

> ### Author Rebuttal · Authors · 2025-03-30
>
> Dear reviewer uVR9, thank you very much for your thorough review. We appreciate that you find our topic “intriguing” and our experimental designs “sound and valid”. In the following we discuss how we have remedied the concerns that you have raised. Your comments have, in our view, considerably strengthened the paper, particularly regarding the inclusion of larger models and the development of a second, more targeted dataset for evaluating between-task generalisation.
> \
> \
> *Testing a broader range of model architectures and including larger models would strengthen the conclusions. Will larger models exhibit different patterns regarding this phenomenon?*
> \
> \
> Thank you for highlighting this. To remedy this weakness, we added a new family consisting of two larger models: Llama-3.2 Vision 11B and 90B. We find the same pattern across both model families and all model sizes - models are capable of generalising to new tower sizes, particularly when trained on shorter towers, but they fail to generalise to naturalistic data or to a new cognitive task (see here for updated versions of Figures 3 and 13 with 7B as reference: [Figure 3](http://postimg.cc/HVcmqHCm), [Figure 13](http://postimg.cc/HjzJNvx9)).
> \
> \
> *It appears that all experiments are conducted on the cubes dataset; I suggest that incorporating other types of physical understanding tasks could also help verify the findings.*
> \
> \
> Thank you for this suggestion. We chose to design the cubeworld dataset to ensure that models have a fair chance of generalizing given that they have become used to the visual characteristics of the environment.  This allows us to infer to some degree whether models’ failure to generalize is due to small differences between the tasks or due to their inability to learn intuitive theories through task-specific fine-tuning. We also designed the tasks in cubeworld based on a long history in cognitive science of using block towers to study physical understanding in humans [1-3]. However, we agree that it is important to be clear about the limitations of our work and have therefore added a sentence to the limitations outlining that we only investigate a subset of intuitive physics here, also taking into account the comments of reviewer mM6z.
> \
> \
> *The experiments do not provide sufficient analysis on the reasons behind the failures in generalization, which limits the insights offered by the study. What are the underlying reasons for the failures in generalization?*
> \
> \
> Thank you for raising this question. To establish whether failures in generalization are due to small differences between tasks, or if the models struggle with learning intuitive theories through task-specific fine-tuning, we added another dataset where differences between the two cognitive domains are kept as minimal as possible. We generate paired images of pyramids, in which the causal reasoning image contains a red block which is removed to generate the intuitive physics image (see this [Figure](http://postimg.cc/3k5Czc3k)).
> \
> \
> In principle, being able to reason about the counterfactual stability of a pyramid ought to predispose models to reason about the factual stability of pyramids. Thus, we expected a transfer from causal reasoning to intuitive physics, especially since we test models using the corresponding images from the pairs they were fine-tuned on. Furthermore, we explicitly tell the models that the red block has been removed. Nevertheless, we do not find evidence of this transfer, suggesting that task-specific fine-tuning does not lead to models learning intuitive theories. Instead, they appear to be learning task-specific superficial shortcuts that do not generalize [4-5].
> \
> \
> *I find that the paper is poorly written and lacks organization. I recommend that the authors reorganize the experimental sections and clarify the conclusions. Figures 3 and 4 are not clearly conveyed, making them difficult to understand.*
> \
> \
> We are sorry to hear this. Conveying our ideas and conclusions clearly is very important to us, so in order to improve this, we added a conclusion for every section of the results. We have also updated the captions of Figures 2, 3, and 4 to make them easier to understand. We hope that these changes improve the readability of our paper.
> \
> \
> *Essential References Not Discussed.*
> \
> \
> Thank you for bringing these references to our attention. We have extended the related works section based on your comments and those of reviewers fjkm and 3bXm.
> \
> \
> \
> [1] Baillargeon, R., & Hanko-Summers, S. (1990). Is the top object adequately supported by the bottom object? Young infants' understanding of support relations.
>
> [2] Spelke, E. S., et al. (1992). Origins of knowledge.
>
> [3] Baillargeon, et al. (1992). The development of young infants' intuitions about support.
>
> [4] Ilyas, A., et al. (2019). Adversarial examples are not bugs, they are features.
>
> [5] Geirhos, R., et al. (2020). Shortcut learning in deep neural networks.

---

> > ### Comment · Reviewer_uVR9 · 2025-04-02
> >
> > I'd like to maintain my current score because of the following reason:
> >
> > - The paper lacks deeper insights, as many of the conclusions are fairly intuitive. It is already well-known in the language modeling community that finetuning often does not lead to substantial improvements in generalization. Moreover, the experimental setup is overly simplistic. Some findings—such as the one in Line 377 stating that finetuning on human judgments increases alignment with human preferences—are fundamental and expected outcomes in machine learning. Essentially, the experiments amount to finetuning the model on different data distributions and evaluating how well those distributions overlap.
> >
> > - The title appears to be overstated. "Reasoning" encompasses a wide range of tasks, including mathematical reasoning, abstract reasoning, and spatial reasoning, among others. The experiments presented in the paper do not sufficiently support such a broad claim.
> >
> > - The study is limited to one finetuning approach—PEFT, specifically QLoRA. If the goal is to explore the limitations of finetuning, the paper should include a broader range of finetuning techniques to substantiate its claims.

---

> > > ### Author Response · Authors · 2025-04-02
> > >
> > > Dear reviewer uVR9, we appreciate your engagement with our paper and understand that you retain a few concerns despite our previous efforts to improve the paper. We want to highlight that based on your initial comments we fine-tuned new models from other families, such as Llama-3.2 Vision 11B and 90B, and added a completely new fine-tuning dataset to better understand when generalization fails – changes we think make our paper stronger. In the following, we address your new set of concerns. We hope that we can convince you that our paper does present novel results that are of interest to the community.
> > > \
> > > \
> > > *The paper lacks deeper insights, as many of the conclusions are fairly intuitive. It is already well-known in the language modeling community that finetuning often does not lead to substantial improvements in generalization. Moreover, the experimental setup is overly simplistic. Some findings—such as the one in Line 377 stating that finetuning on human judgments increases alignment with human preferences—are fundamental and expected outcomes in machine learning. Essentially, the experiments amount to finetuning the model on different data distributions and evaluating how well those distributions overlap.*
> > > \
> > > \
> > > Thank you for this comment. While we are aware of concurrent works showcasing fine-tuned models’ difficulty generalizing [1], we want to stress that we are interested in fine-tuning for improving visual cognitive abilities, a topic that has not been investigated before.
> > > \
> > > \
> > > We agree that the experimental setup is simplistic. This is on purpose and allows us to test in which cases fine-tuned VLMs generalize and in which cases they don’t. However, we do not think our results are obvious: we did not expect VLMs to generalize at all – however, we find that they perform well on smaller or larger towers from the task they are fine-tuned on. Thus, our results do not show that generalization does not occur at all, but rather that it is limited to the specific task at hand.
> > > \
> > > \
> > > We were also surprised that this task specificity extends to cases where the visual stimuli are almost identical. Insofar we do not agree that our experiments “amount to fine-tuning the model on different data distributions and evaluating how well those distributions overlap.” The new experiment we added in response to your first set of comments shows that fine-tuned VLMs do not generalize even between very similar data distributions. Here, we fine-tune VLMs on block pyramids with a single red block and ask them whether any other block would fall if the red block was removed. After fine-tuning they can do this very well. But given the same images they saw during fine-tuning, only without the red block, they fail to reason about the factual stability of the tower (see postimg.cc/3k5Czc3k) – even if we explicitly tell them that the red block was removed to help them make the connection between counterfactual and factual stability. This is very interesting and unexpected, because reasoning about counterfactual stability should require reasoning about factual stability. We think this result is of importance to the community, as it suggests that VLMs do not learn robust visual cognitive abilities during fine-tuning, but rather that they rely on task specific shortcuts that do not generalize between tasks.
> > > \
> > > \
> > > *The title appears to be overstated. "Reasoning" encompasses a wide range of tasks, including mathematical reasoning, abstract reasoning, and spatial reasoning, among others. The experiments presented in the paper do not sufficiently support such a broad claim.*
> > > \
> > > \
> > > Thank you for highlighting this. We agree that reasoning is a broad term that incorporates a number of different types of reasoning. We here are interested in visual cognitive reasoning and we investigate tasks that are representative for this type of reasoning. We understand that the title could potentially be misleading. We are happy to change it to the more specific “Testing the limits of supervised fine-tuning to improve visual cognition in vision language models”.
> > > \
> > > \
> > > *The study is limited to one finetuning approach—PEFT, specifically QLoRA. If the goal is to explore the limitations of finetuning, the paper should include a broader range of finetuning techniques to substantiate its claims.*
> > > \
> > > \
> > > We are first and foremost interested in fine-tuning for improving visual cognition. We chose QLoRA as it is a widespread and efficient method, and we think it allows for some generalizable insights into establishing the limits of supervised fine-tuning for improving visual cognition in vision language models. To be more specific about our objective, we propose to change the title to this more specific title as outlined above. We also added clarification on the exact type of fine-tuning used to the introduction and highlighted this limitation in the discussion.
> > > \
> > > \
> > > [1] Chu, Tianzhe, et al. "SFT memorizes, RL generalizes: A comparative study of foundation model post-training."

---

### Official Review · Reviewer_mM6z · 2025-03-14

**Overall Recommendation:** 3

**Summary:**

This is an interesting piece of work that seems to be among the first to investigate the following: fine-tuning is a widespread approach to improving LLM performance in the text domain, but for domains such as intuitive physics and causal reasoning, which are not text-related and not really purely visual capabilties either, how can can finetuning go to develop or improve such fundamental capabilities in VLMs?

One would not really expect this to work extremely well (unlike text), and I doube the author expected it either, but the paper does shed some light on the extent to which it works. In short, it works somewhat well within the domain (not that surprisingly), but does not generalize to the other domain.

## Update after rebuttal:

I find that the work was/is generally interesting, provides a useful contribution, and perhaps worth publicizing (i.e. acceptance).

My main issues were with the lack of robustness in the evaluation, and the over-generality of the claims, esp. the title. At the very least, the work is not about "reasoning" in general.

Critically contingent on the new expt results (more model, repeats, etc.) claimed by the authors, I upgrade my review score from 2 to 3.

However, I would strongly, strongly recommend that the title needs to revised and scoped down. (Whether the AC/SAC can enforce that, I'm not sure...)

**Claims And Evidence:**

1) Whatever claims the paper makes, they are severely limited by a number of factors, as already pointed out by the authors themselves in the Discussion.

2) More importantly though, intuitive physics and causal reasoning are very broad capabilities, and the blocksworld-like datasets used here are a very small sliver of these domains/capabilities. I'm not sure any claim about intuitive physics whatsoever is supported. Rather, any claims should be minimally be scoped down to the very specific thing being investigated, e.g. stability, etc. In addition, these need to be caveated that only solid/rigid cubes of uniform density are used, etc. etc.

**Essential References Not Discussed:**

Nil.

**Experimental Designs Or Analyses:**

Experimental designs and analyses were generally sound, but subject to a number of limitations, some of which the authors themselves point out (e.g. size of models finetuned, alternative finetuning procedures, etc. etc.)

**Methods And Evaluation Criteria:**

Methods and evaluation criteria were generally sound.

**Other Comments Or Suggestions:**

Nil.

**Other Strengths And Weaknesses:**

Nil.

**Questions For Authors:**

Nil.

**Relation To Broader Scientific Literature:**

This paper is generally somewaht useful in terms of contributing some knowledge about what works (or not) to what extent for improving intuitive physics and causal reasoning through fine-tuning of VLMs. However, as the authors themselves point out in the Discussion section, there are a number of limitations, which in my view quite significantly limit the usefull of this paper.

**Theoretical Claims:**

Not applicable.

---

> ### Author Rebuttal · Authors · 2025-03-30
>
> Dear reviewer mM6z, thank you very much for your feedback. We appreciate that you think our work is interesting and that the “designs and analyses were generally sound”. Furthermore, we are glad to hear that you agree that our work “does shed some light on the extent to which [vision finetuning] works”. Below, we discuss the specific concerns you raised and the revisions we have made, which we think have strengthened our paper.
> \
> \
> *One would not really expect [fine-tuning to improve fundamental capabilities in VLMs] to work extremely well (unlike text), and I doubt the authors expected it either.*
> \
> \
> Thank you for this comment. As you suggest, finetuning has proven to be an effective method for producing generalisable behaviour in text-only models. However, as also highlighted by reviewer fjkm, we are not aware of prior work that establishes the link between fine-tuning and improved visual cognition in VLMs. Therefore, our work here seeks to investigate whether finetuning can confer an advantage to the visual domain too.
> \
> \
> While previous work has established that pre-trained VLMs do not perform well in visual cognition tasks, we did not know whether fine-tuning could sufficiently improve their performance on these tasks. We were surprised to find that fine-tuning can make models perform well on the specific task they are fine-tuned on. Furthermore, we found that the models can even generalize robustly to unseen towers of sizes that they had not seen during training. Both of these results show that fine-tuning works surprisingly well in improving VLM performance on specific tasks and domains.
> \
> \
> However, the ways in which models can generalize from their fine-tuning data is severely limited. We show that they have trouble generalizing to visually distinct stimuli as well as to visually similar stimuli in another domain. The take-away from this investigation therefore should not be that fine-tuning does not improve VLM capabilities, but rather that it leads to very task-specific improvements that do not generalize well.
> \
> \
> *Whatever claims the paper makes, they are severely limited by a number of factors, as already pointed out by the authors themselves in the Discussion.*
> \
> \
> The main limitations we outline in the discussion were that we: (1) investigate only smaller models of a single model family, and (2) use a single parameterisation for fine-tuning and only one dataset distribution per domain. We have now fixed these limitations by: (1) adding bigger models of another class with 11B and 90B parameter versions of Llama-3.2 Vision; (2) conducting three repetitions of every model on every dataset, using a different adapter weight initialisation and a different sample of fine-tuning data. Our new results follow the same pattern as our previous results. We think that these extensions have significantly improved the generality and robustness of our claims.
> \
> \
> *I'm not sure any claim about intuitive physics whatsoever is supported. Rather, any claims should be minimally scoped down to the very specific thing being investigated.*
> \
> \
> We agree that intuitive physics refers to a broad set of capabilities, of which we only investigate a subset. The tower stacking task is a canonical task that has long been used as a testbed for intuitive physical capabilities in machine learning systems [1-4], drawing on an even longer history in cognitive science using the same task [5-7]. However, we also agree that it is important to be specific about the scope of the investigated capabilities. We have therefore outlined the scope of our experiments more specifically in the discussion section, namely, that we focus on model intuitions for stability judgements involving solid, uniformly dense blocks.
> \
> \
> [1] Lake, B. M., Ullman, T. D., Tenenbaum, J. B., & Gershman, S. J. (2017). Building machines that learn and think like people.
>
> [2] Battaglia, P., Pascanu, R., Lai, M., & Jimenez Rezende, D. (2016). Interaction networks for learning about objects, relations and physics.
>
> [3] Lerer, A., Gross, S., & Fergus, R. (2016). Learning physical intuition of block towers by example.
>
> [4] Piloto, L. S., Weinstein, A., Battaglia, P., & Botvinick, M. (2022). Intuitive physics learning in a deep-learning model inspired by developmental psychology.
>
> [5] Baillargeon, R., & Hanko-Summers, S. (1990). Is the top object adequately supported by the bottom object? Young infants' understanding of support relations.
>
> [6] Spelke, E. S., Breinlinger, K., Macomber, J., & Jacobson, K. (1992). Origins of knowledge.
>
> [7] Baillargeon, R., Needham, A., & DeVos, J. (1992). The development of young infants' intuitions about support.

---

### Official Review · Reviewer_fjkm · 2025-03-15

**Overall Recommendation:** 4

**Summary:**

This paper explores the limitations of Vision-Language Models (VLMs) in causal understanding of the physical world — a problem that is quite interesting to the community. The authors examine the potential of fine-tuning (FT) as a method to improve performance on intuitive physics and causal reasoning tasks. They fine-tune a VLM on tasks from the cognitive domain, specifically intuitive physics and causal reasoning in CubeWorld and a real-world environment. However, the results indicate that fine-tuning alone does not achieve the desired abilities. The model is evaluated on tasks such as IID stability of towers, OOD generalization with different numbers of blocks, OOD domain transfer from CubeWorld to real-world scenes, and counterfactual tasks.

**Claims And Evidence:**

Yes, but a more rigorous study could be done as pointed out in the "Cons" below.

**Essential References Not Discussed:**

Essential references appear to be cited.

**Experimental Designs Or Analyses:**

Yes. I don't have any issues with the design/analyses currently in the paper -- except that there could be a more complete study as discussed in the Cons below.

**Methods And Evaluation Criteria:**

Yes, the benchmark, method, and evaluation criteria make sense. I am a bit unclear about why the alignment with human behavior is of importance that the paper reports.

**Other Comments Or Suggestions:**

See Cons listed above.

**Other Strengths And Weaknesses:**

## Pros

Very interesting findings:

1. Fine-tuning on CubeWorld does not lead to good performance on real world data suggesting the model / fine-tuning process is lacking the ability to work with abstractions.
2. The ability to solve one task does not lend it the ability to solve a different yet related task.

## Cons

1. The 2B and 7B might be too small. I don’t mean to say that a larger model would necessarily resolve all problems, just that 2B and 7B may be too small to claim it as the “limits of fine-tuning”.
2. Also, having only Qwen in the evaluation is a limitation — having at least one other class of VLM in the evaluation would provide a better perspective. Else, the results might be too specific to Qwen and don’t provide a general message to the community.
3. The related work seems too small and could be expanded to include studies that discuss (1) causal understanding/generalization on visual scenes in the pre-VLM era, and (2) Any other relevant VLM works and how they leave a gap with regards to evaluating causal understanding ability. (3) Studies on VLMs focusing on other kinds of reasoning abilities (not necessarily causal understanding/reasoning or perhaps using text only).

**Questions For Authors:**

> Alignment with human behavior
>

What is the rationale behind seeking alignment with human behavior? (I don’t mean to argue against it, just seeking more clarification.)

> L407 (right): Models’ inability to generalize to another cognitive domain is not due to them being limited in parameters or potential ability — models fine-tuned on a mixture of intuitive physics and causal reasoning data performed well in both domains.
>

Yes, but fine-tuning and how weights encode sub-routines for solving the task can be affected by the number of parameters. Wonder if the authors have thoughts about this and if it would be informative to show whether scaling model size has any effect on the generalization / causal understanding performance. The authors later point out in the discussion that this is a future work.

**Relation To Broader Scientific Literature:**

To me, it seems that evaluation studies about the limitations of VLMs are only gradually emerging. Previous studies have not explored the intersection of visual causal understanding, fine-tuning, and VLMs.

**Theoretical Claims:**

Not applicable.

---

> ### Author Rebuttal · Authors · 2025-03-30
>
> Dear reviewer fjkm, thank you very much for your comments. We appreciate that you think the findings are “very interesting” and that you agree that our “method and evaluation criteria make sense”. In the following we discuss the specific concerns that you raised and how we have sought to remedy them. Your comments have led to our paper becoming considerably stronger, particularly with the inclusion of larger models and the clarification of key methodological points.
> \
> \
> *The 2B and 7B might be too small and having at least one other class of VLM in the evaluation would provide a better perspective. Fine-tuning and how weights encode sub-routines for solving the task can be affected by the number of parameters.*
> \
> \
> Thank you for raising this point. We agree that testing only the 2B and 7B Qwen models restricts the generality of the claims we can make based on our experiments. To remedy this, we have added two bigger models from another model family: Llama-3.2 Vision 11B and 90B. We find the same pattern across both model classes and all model sizes - models are capable of generalising to new tower sizes, particularly when trained on shorter towers, but they fail to generalise to naturalistic data or to a new cognitive task (see here for updated versions of Figures 3 and 13 with 7B as reference: [Figure 3](http://postimg.cc/HVcmqHCm), [Figure 13](http://postimg.cc/HjzJNvx9)).
> \
> \
> *The related work seems too small and could be expanded to include studies that discuss (1) causal understanding/generalization on visual scenes in the pre-VLM era, and (2) Any other relevant VLM works and how they leave a gap with regards to evaluating causal understanding ability. (3) Studies on VLMs focusing on other kinds of reasoning abilities (not necessarily causal understanding/reasoning or perhaps using text only).*
> \
> \
> Thank you for this comment, which also echoes the comments of reviewers uVR9 and 3bXm. Regarding point (1), we have referenced the CLEVRER dataset and a prominent model that is trained on it [1-3], which are key works studying causal reasoning and generalization in computer vision systems. Regarding points (2) and (3), we have referenced several papers suggested by the other reviewers. In particular, we have included references to other papers discussing causal reasoning and generalisation on other cognitive tasks in LLMs and VLMs [4-7]. We have also included references to works proposing explanations for why VLMs may struggle to generalise to tasks such as intuitive physics and causal reasoning [8-9], as well as work employing more varied fine-tuning datasets to improve performance [10].
> \
> \
> *What is the rationale behind seeking alignment with human behavior?*
> \
> \
> Thank you for this question. We realize that we were not explicit enough about the hypotheses behind seeking alignment with human behavior. We have added further explanation to the related works section, including [10-11], which were also pointed out by reviewer 3bXm. Binz et al. show that fine-tuning on human choices can lead to models that predict human behavior in previously unseen tasks. On our tasks, human choices and the ground truth are not perfectly aligned, and we sought to explore whether fine-tuning could (a) align models with human choices, and (b) whether training on human choices would lead to improved generalisation performance. We were interested in exploring (b) because fine-tuning on human choices could lead to models learning human intuitions, which might be more robust and generalizable. We confirmed (a) but found only limited evidence for (b): fine-tuning on human behaviour only confers a slight advantage at transferring to the naturalistic Lerer et al. (2016) tower blocks, but no detectable advantage for transferring to a different cognitive task.
> \
> \
> \
> [1] Johnson, J.,  et al. (2017). Clevr: A diagnostic dataset for compositional language and elementary visual reasoning.
>
> [2] Yi, Kexin, et al. (2020). Clevrer: Collision events for video representation and reasoning.
>
> [3] Chen, Zhenfang, et al. (2021). Grounding physical concepts of objects and events through dynamic visual reasoning.
>
> [4] Dziri, et al.  (2023). Faith and fate: Limits of transformers on compositionality.
>
> [5] Zhou, H., et al. (2023). What algorithms can transformers learn? A study in length generalization.
>
> [6] Zhou, R., et al. (2024). Math for AI: On the Generalization of Learning Mathematical Problem Solving.
>
> [7] Li, C., et al.  (2024). In-context compositional generalization for large vision-language models.
>
> [8] Campbell, D., et al. (2024). Understanding the limits of vision language models through the lens of the binding problem.
>
> [9] Frankland, S. M. et al. (2025). No Coincidence, George: Processing Limits in Cognitive Function Reflect the Curse of Generalization.
>
> [10] Binz, M. et al.  (2024). Centaur: a foundation model of human cognition.
>
> [11] Binz, M. & Schulz, E. (2023). Turning large language models into cognitive models.

---

> > ### Comment · Reviewer_fjkm · 2025-04-06
> >
> > Thank you for the response. I have revised my score.

---

### Decision · Program_Chairs · 2025-05-01

**Decision:**

Accept (poster)

**Comment:**

This study aims to enhance the visual cognition of VLMs by incorporating visual stimuli and human feedback. Specifically, the authors fine-tune the model on intuitive physics and causal reasoning tasks. While all reviewers agree that the study addresses an interesting research question, they also raise concerns regarding its applicability and experimental validity. Overall, the strengths outweigh the weaknesses, and I am inclined to accept this paper.